# A Unified Algorithm Framework for Unsupervised Discovery of Skills based on Determinantal Point Process

**Jiayu Chen**
Purdue University
West Lafayette, IN 47907
chen3686@purdue.edu

**Vaneet Aggarwal**
Purdue University
West Lafayette, IN 47907
vaneet@purdue.edu

**Tian Lan**
The George Washington University
Washington, DC 20052
tlan@gwu.edu

## Abstract

Learning rich skills under the option framework without supervision of external rewards is at the frontier of reinforcement learning research. Existing works mainly fall into two distinctive categories: variational option discovery that maximizes the diversity of the options through a mutual information loss (while ignoring coverage) and Laplacian-based methods that focus on improving the coverage of options by increasing connectivity of the state space (while ignoring diversity). In this paper, we show that diversity and coverage in unsupervised option discovery can indeed be unified under the same mathematical framework. To be specific, we explicitly quantify the diversity and coverage of the learned options through a novel use of Determinantal Point Process (DPP) and optimize these objectives to discover options with both superior diversity and coverage. Our proposed algorithm, ODPP, has undergone extensive evaluation on challenging tasks created with Mujoco and Atari. The results demonstrate that our algorithm outperforms state-of-the-art baselines in both diversity- and coverage-driven categories.

## 1 Introduction

Reinforcement Learning (RL) has achieved impressive performance in a variety of scenarios, such as games [1, 2], robotic control [3, 4], and transportation [5, 6]. However, most of its applications rely on carefully-crafted, task-specific rewards to drive exploration and learning, limiting its use in real-life scenarios often with sparse or no rewards. To this end, utilizing unsupervised option discovery – acquiring rich skills without supervision of environmental rewards by building temporal action-sequence abstractions (denoted as options), to support efficient learning can be essential. The acquired skills are not specific to a single task and thus can be utilized to solve multiple downstream tasks by implementing a corresponding meta-controller that operates hierarchically on these skills.

Existing unsupervised option discovery approaches broadly fall into two categories: (1) Variational option discovery, e.g., [7, 8, 9], which aims to improve diversity of discovered options by maximizing the mutual information [10] between the options and trajectories they generate. It tends to reinforce already discovered behaviors for improved diversity rather than exploring (e.g., visiting poorly-connected states) to discover new ones, so the learned options may have limited coverage of the state space. (2) Laplacian-based option discovery, e.g., [11, 12], which clusters the state space using a Laplacian spectrum embedding of the state transition graph, and then learns options to connect different clusters. This approach is shown to improve the algebraic connectivity of the state space [13] and reduce expected covering time during exploration. However, the discovered options focus on improving connectivity between certain clusters and thus could be homogeneous and lack diversity. Note that coverage in this paper is defined as a property with respect to a single option, which can be measured as the number of state clusters traversed by an option trajectory. By maximizing coverage

37th Conference on Neural Information Processing Systems (NeurIPS 2023).

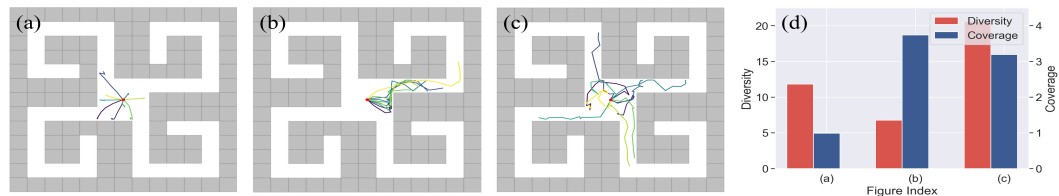

Figure 1: Illustrative example. (a) Options from variational methods have good diversity but poor (single-option) coverage of the corridors. (b) Each option from Laplacian-based methods aims to improve its coverage by visiting poorly-connected corner states but tends to be homogeneous as the others. (c) Options from our proposed method have both superior diversity and coverage. (d) The coverage and diversity measure of the options in (a)-(c) defined with DPP (i.e., Eq.(7) and (9)).

of each single option and diversity among different options in the meanwhile, the overall span of all options can be maximized. However, diversity and coverage may not go hand-in-hand in option discovery, as visualized in Figure 1(a)(b). Attempts such as [14, 15] have been made to address this gap, but they rely on expert datasets that contain diverse trajectories spanning the whole state spaces and lack an analytical framework for diversity and coverage of the discovered options.

This paper introduces a novel framework for unsupervised option discovery by utilizing Determinantal Point Process (DPP) to quantify and optimize both diversity and (single-option) coverage of the learned options. A DPP establishes a probability measure for subsets of a set of items. The expected cardinality, representing the average size of random subsets drawn according to the DPP, serves as a diversity measure, as it reflects the likelihood of sampling diverse items and the number of distinct items in the set. **First**, to enhance option diversity, we apply a DPP to the set of trajectories for different options. Maximizing the expected cardinality under this DPP encourages agents to explore diverse trajectories under various options. **Second**, we create another DPP for the set of visited states in a trajectory and maximize its expected cardinality. This prompts the agent to visit distant states from distinct state clusters through each trajectory, leading to higher single-option coverage. As shown in Figure 1(d), these objectives effectively measure diversity and coverage of the options. **Lastly**, to establish the option-policy mapping and so learn multiple options simultaneously, we maximize the mutual information between the option choice and related trajectories, as in variational option discovery. Rather than using the whole trajectory [16] or only the goal state [7], our solution extract critical landmark states from a trajectory via a maximum a posteriori (MAP) inference of a DPP, allowing noise mitigation while retaining vital information. **In conclusion**, our proposed framework unifies diversity and coverage in option discovery using a mathematical modeling tool—DPP. Our **key contributions** are as follows. (1) To the best of our knowledge, this is the first work to adopt DPP for option discovery. (2) The proposed unified framework enables explicit maximization of option diversity and coverage, capturing advantages of both variational and Laplacian-based methods. (3) Empirical results on a series of challenging RL tasks demonstrates the superiority of our algorithm over state-of-the-art (SOTA) baselines.

## 2 Background and Related Works

### 2.1 Unsupervised Option Discovery

As proposed in [17], the option framework consists of three components: an intra-option policy $\pi : \mathcal{S} \times \mathcal{A} \to [0, 1]$, a termination condition $\beta : \mathcal{S} \to \{0, 1\}$, and an initiation set $I \subseteq \mathcal{S}$. An option $< I, \pi, \beta >$ is available in state $s$ if and only if $s \in I$. If the option is taken, actions are selected according to $\pi$ until it terminates according to $\beta$ (i.e., $\beta = 1$). The option framework enables learning and planning at multiple temporal levels and has been widely adopted in RL. Multiple research areas centered on this framework have been developed. Unsupervised Option Discovery aims at discovering skills that are diverse and efficient for downstream task learning without supervision from rewards, for which algorithms have been proposed for both single-agent and multi-agent scenarios [18, 19, 20, 21]. Hierarchical Reinforcement Learning [22, 23] and Hierarchical Imitation Learning [24, 25, 26], on the other hand, aim at directly learning a hierarchical policy incorporated with skills, either from interactions with the environment or expert demonstrations.

We focus on single-agent unsupervised option discovery. One primary research direction in this field is variational option discovery, which aims to learn a latent-conditioned option policy $\pi(a|s, c)$, where $c$ represents the option latent. Diverse options can be learned by maximizing the mutual information between the option choice $c$ and the trajectory generated by the policy conditioned on $c$. As shown in [27], this objective is equivalent to a Variational Autoencoder (VAE) [28]. Variational option discovery can be categorized into two groups based on the structure of the VAE used: **(1)** Trajectory-first methods, such as EDL [14] and OPAL [15], follow the structure $\tau \xrightarrow{E} c \xrightarrow{D} \tau$. Here, $\tau$ represents the agent's trajectory, while $D$ and $E$ denote the VAE's decoder and encoder, respectively. In this case, the policy network $\pi(a|s, c)$ serves as the decoder $D$. The effectiveness of these methods rely on the quality of the expert trajectory set used as the encoder's input. For instance, OPAL employed for Offline RL [29] assumes access to a trajectory dataset generated by a mix of diverse policies starting from various initial states. Similarly, in EDL, highly-efficient exploration strategies such as State Marginal Matching [16] are utilized to simulate perfect exploration and obtain a set of trajectories ($\tau$) with good state space coverage for option learning. **(2)** Option-first methods, such as VIC [7], DIAYN [8], VALOR [27], and DADS [9], follow the structure $c \xrightarrow{E} \tau \xrightarrow{D} c$ and learn $\pi(a|s, c)$ as the encoder $E$. Unlike EDL and OPAL learning from "exploration experts", these methods start from a random policy to discover diverse options. However, due to challenges in exploration, they fail to expand the set of states visited by the random policy, leading to poor state space coverage. For instance, in Figure 1(a), though trajectories of different options are diverse, none of them explore the corridor. There have been more notable advancements in variational option discovery methods, as cited in [30, 31, 32, 33, 34, 35]. A detailed comparison between these methods and our algorithm, highlighting our contributions, can be found in Appendix B.

In this paper, we tackle a more challenging scenario where the agent must learn to identify diverse options and thoroughly explore the environment, starting from a random policy. This approach supports self-driven learning and exploration, rather than relying on expert policy/trajectories like trajectory-first methods. We achieve this by integrating the variational method with another key branch in unsupervised option discovery—Laplacian-based option discovery [11, 12]. This approach is based on the Laplacian spectrum of the state transition graph, which can be estimated using state transitions in the replay buffer. The state transition graph and its Laplacian matrix are formally defined in Appendix A.1. Specifically, the approach presented in [12] first estimates the Fiedler vector – the eigenvector associated with the second smallest eigenvalue of the Laplacian. Options are then trained to connect states with high and low values in the Fiedler vector. These states are loosely connected by "bottleneck" states. By connecting them with options, the algebraic connectivity of the state space is enhanced, and thus the exploration can be accelerated [13]. However, these options may be homogeneous. For instance, in Figure 1(b), multiple options are found to explore the right corridor, but they all follow the same direction given by the Fiedler vector. Inspired by variational and Laplacian-based methods, we propose a new option discovery framework that unifies the optimization of diversity and coverage, through a novel use of DPP. As shown in Figure 1(c), multiple diverse options are discovered, most of which traverse the "bottleneck" states to enhance coverage.

## 2.2 Determinantal Point Process

According to [36], given a set of items $\mathcal{W} = \{w_1, \cdots, w_N\}$, a point process $\mathcal{P}$ on $\mathcal{W}$ is a probability measure on the set of all the subsets of $\mathcal{W}$. $\mathcal{P}$ is called a Determinantal Point Process (DPP) if a random subset $\mathbf{W}$ drawn according to $\mathcal{P}$ has probability:

$$\mathcal{P}_{L(\mathcal{W})}(\mathbf{W} = W) = \frac{\det(L_W)}{\sum_{W' \subseteq \mathcal{W}} \det(L_{W'})} = \frac{\det(L_W)}{\det(L + I)} \tag{1}$$

$I \in \mathbb{R}^{N \times N}$ is the identity matrix. $L = L(\mathcal{W}) \in \mathbb{R}^{N \times N}$ is the DPP kernel matrix, which should be symmetric and positive semidefinite. $L_W \in \mathbb{R}^{|W| \times |W|}$ is the sub-matrix of $L$ indexed by elements in $W$. Specifically, $P_L(\mathbf{W} = \{w_i\}) \propto L_{ii}$ and $P_L(\mathbf{W} = \{w_i, w_j\}) \propto L_{ii}L_{jj} - L_{ij}^2$ where $L_{ij}$ measures the similarity between item $i$ and $j$. Since the inclusion of one item reduces the probability of including similar items, sets consisting of diverse items are more likely to be sampled by a DPP.

The DPP kernel matrix $L$ can be constructed as a Gram Matrix [37]: $L = \widetilde{B}^T \widetilde{B} = Diag(\overrightarrow{q}) \cdot B^T B \cdot Diag(\overrightarrow{q})$. $\overrightarrow{q} = [q_1, \cdots, q_N] \in \mathbb{R}^N$ with $q_i \geq 0$ denotes the quality measure. $B = \left[ \overrightarrow{b_1} \cdots \overrightarrow{b_N} \right] \in \mathbb{R}^{D \times N}$ is the stacked feature matrix where $\overrightarrow{b_i} \in \mathbb{R}^D$ is the feature vector corresponding to $w_i$. The

inner product of feature vectors, e.g., $\vec{b_i}^T \vec{b_j}$, is used as the similarity measure between items in $\mathcal{W}$. From Eq. (1), we can see that $\mathcal{P}_L(\mathbf{W} = W)$ is propotional to the squared $|W|$-dimension volume of the parallelepiped spanned by the columns of $\widetilde{B}$ corresponding to the elements in $W$. Diverse sets have feature vectors that are more orthogonal and span larger volumes, making them more probable.

The expected cardinality of samples from a DPP is an effective measure of the diversity of $\mathcal{W}$ and reflects the number of modes in $\mathcal{W}$ [38]. We provide detailed reasons why we choose the expected cardinality instead of the likelihood in Eq. (1) as the diversity measure in Appendix C.1. According to [36], the expected cardinality of a set sampled from a DPP can be calculated with Eq. (2), where $\lambda_i^{\mathcal{W}}$ is the $i$-th eigenvalue of $L(\mathcal{W})$. DPPs have found applications across a wide array of domains to promote diversity due to their unique ability to model diverse subsets, such as information retrieval [39, 40], computer vision [38], natural language processing [41, 42], and reinforcement learning [43, 44, 45]. This motivates us to employ DPPs in skill discovery for diversity enhancement.

$$\mathbb{E}_{\mathbf{W} \sim P_{L(\mathcal{W})}} [|\mathbf{W}|] = \sum_{i=1}^{N} \frac{\lambda_i^{\mathcal{W}}}{\lambda_i^{\mathcal{W}} + 1} \tag{2}$$

## 3 Proposed Approach

In this section, we propose a DPP-based framework that unifies the variational and Laplacian-based option discovery to get options with both superior diversity and coverage. As discussed in Section 2.1, defining an option requires relating it to its initial states $s_0$, specifying its termination condition, and learning the intra-option policy. Our algorithm learns a prior network $P_\omega(c|s_0)$ to determine the option choice $c \in \mathcal{C}$ at an initial state, and an intra-option policy network $\pi_\theta(a|s, c)$ to interact with the environment using a certain option for a fixed number of steps (i.e., the termination condition). With $P_\omega$ and $\pi_\theta$, we can collect trajectories $\tau = (s_0, a_0, \cdots, s_T)$ corresponding to different options.

In Section 3.1, we propose $\mathcal{L}^{IB}$, a lower bound for the mutual information between the option and the landmark states in its trajectory, inspired by variational option discovery. By introducing the conditional variable $c$ and maximizing $\mathcal{L}^{IB}$, we can establish the option-policy mapping and learn multiple options simultaneously. Each option can generate specific trajectories. However, the mutual information objective only implicitly measures option diversity as the difficulty of distinguishing them via the variational decoder, and does not model the coverage. Thus, in Section 3.2, we additionally introduce explicit coverage and diversity measures based on DPP as objectives. In particular, $\mathcal{L}_1^{DPP}$ measures single-agent coverage as the number of landmark states traversed by an option trajectory. Maximizing this metric encourages each option to cover a broader area in the state space. Notably, $\mathcal{L}_1^{DPP}$ generalizes the Laplacian-based option discovery objectives by employing the Laplacian spectrum as the feature vector to construct the DPP kernel matrix. $\mathcal{L}_2^{DPP}$ measures the diversity among trajectories of the same option. By minimizing it, the consistency these trajectories can be enhanced. $\mathcal{L}_3^{DPP}$ measures the diversity among trajectories corresponding to different options, which should be maximized to improve the diversity among various options. The rationale behind introducing $\mathcal{L}_{1:3}^{DPP}$ is to maximize both the coverage of each individual option and the diversity among various options. By doing so, we can maximize the overall span of all options, thereby fostering the formation of diverse skills that fully explore the state space.

### 3.1 Landmark-based Mutual Information Maximization

Previous works tried to improve the diversity of learned options by maximizing the mutual information between the option choice $c$ and trajectory $\tau$ [27] or goal state $s_T$ [46] generated by the corresponding intra-option policy. However, the whole trajectory contains noise and the goal state only cannot sufficiently represent the option policy. In this paper, we propose to maximize the mutual information between $c$ and landmark states $G$ in $\tau$ instead. $G$ is a set of distinct, representative states. Specifically, after clustering all states according to their features, a diverse set of landmarks with varied feature embeddings is identified to represent each different cluster. Notably, $G$ can be extracted from $\tau$ through the maximum a posteriori (MAP) inference of a DPP [36], shown as Eq. (3) where $\mathcal{X}$ denotes the set of states in $\tau$. The intuition is that these landmarks should constitute the most diverse subset of $\mathcal{X}$ and thus should be the most probable under this DPP.

$$G = \arg\max_{X \subseteq \mathcal{X}} \mathcal{P}_{L(\mathcal{X})}(\mathbf{X} = X) = \arg\max_{X \subseteq \mathcal{X}} \mathcal{P}(\mathbf{X} = X | \mathbf{L} = L(\mathcal{X})) = \arg\max_{X \subseteq \mathcal{X}} \frac{\det(L_X)}{\det(L + I)} \tag{3}$$

In order to maximize the mutual information between $G$ and $c$ while filtering out the redundant information for option discovery in $\tau$. According to the Information Bottleneck framework [47], this can be realized through Eq. (4), where $\mu(\cdot)$ is the initial state distribution, $I(\cdot)$ denotes the mutual information, and $I_{ct}$ is the information constraint.

$$\max_{\theta,\omega} \mathbb{E}_{s_0 \sim \mu(\cdot)} I(c, G|s_0; \theta, \omega), \ s.t., \ I(c, \tau|s_0; \theta, \omega) \le I_{ct} \tag{4}$$

Equivalently, with the introduction of a Lagrange multiplier $\beta \ge 0$, we can optimize:

$$\max_{\theta,\omega} \mathbb{E}_{s_0 \sim \mu(\cdot)} [I(c, G|s_0; \theta, \omega) - \beta I(c, \tau|s_0; \theta, \omega)] \tag{5}$$

It is infeasible to directly compute and optimize Eq. (5), so we have the following proposition.

**Proposition 1.** *The optimization problem as Eq. (5) can be solved by maximizing $\mathcal{L}^{IB}(\omega, \theta, \phi)$:*

$$\mathcal{H}(\mathcal{C}|\mathcal{S}) + \mathbb{E}_{s_0,c,\tau} \left[ P^{DPP}(G|\tau) \log P_\phi(c|s_0, G) \right] - \beta \mathbb{E}_{s_0,c} [D_{KL}(P_\theta(\tau|s_0, c)||Unif(\tau|s_0))] \tag{6}$$

*Here, $s_0 \sim \mu(\cdot), c \sim P_\omega(\cdot|s_0), \tau \sim P_\theta(\cdot|s_0, c)$. $\mathcal{H}(\mathcal{C}|\mathcal{S})$ represents the entropy associated with the option choice distribution at a given state. $P^{DPP}(G|\tau) = \det(L_G)/\det(L + I)$ is the probability of extracting $G$ from $\tau$, under a DPP on the set of states in $\tau$. $P_\phi(c|s_0, G)$ serves as a variational estimation of the posterior term $P(c|s_0, G)$ in $I(c, G|s_0)$. $Unif(\tau|s_0)$ denotes the probability of sampling trajectory $\tau$ given $s_0$ under a uniformly random walk policy.*

The derivation can be found in Appendix A.2. **(1)** The first two terms of Eq. (6) constitute a lower bound of the first term in Eq. (5). $\mathcal{H}(\mathcal{C}|\mathcal{S})$ and $P_\phi(c|s_0, G)$ can be estimated using the output from the prior network and variational posterior network, respectively. **(2)** The third term in Eq. (6) corresponds to a lower bound of the second term in Eq. (5). Instead of directly calculating $-\beta I(c, \tau|s_0)$ which is implausible, we introduce $Unif(\tau|s_0)$ to convert it to a regularization term as in Eq. (6). Specifically, by minimizing $D_{KL}(P_\theta(\tau|s_0, c)||Unif(\tau|s_0))$, which is the KL Divergence [10] between the distribution of trajectories under our policy and a random walk policy, exploration of the trained policy $\pi_\theta$ can be improved. Note that $P_\theta(\tau|s_0, c) = \prod_{t=0}^{T-1} \pi_\theta(a_t|s_t, c)P(s_{t+1}|s_t, a_t)$, where $P(s_{t+1}|s_t, a_t)$ is the transition function in the MDP.

Another challenge in calculating $\mathcal{L}^{IB}$ is to infer the landmarks $G$ with Eq. (3). ($L$ is the kernel matrix of the DPP, of which the construction is further introduced in the next section.) The MAP inference problem related to a DPP is NP-hard [48], and greedy algorithms have shown to be promising as a solution. In this work, we adopt the fast greedy method proposed in [39] for the MAP inference, which is further introduced in Appendix C.2 with its pseudo code and complexity analysis.

## 3.2 Quantifying Diversity and Coverage via DPP

In this section, we propose three optimization terms defined with DPP to explicitly model the coverage and diversity of the learned options. Jointly optimizing these terms with Eq. (6), the discovered options are expected to exhibit improved state space coverage and enhanced diversity.

The first term relates to improving the coverage of each option by maximizing:

$$\mathcal{L}_1^{DPP}(\omega, \theta) = \mathbb{E}_{s_0} \left[ \sum_{c,\tau} P_\omega(c|s_0)P_\theta(\tau|s_0, c)f(\tau) \right], \ f(\tau) = \mathbb{E}_{\mathbf{X} \sim P_{L(\mathcal{X})}} [|\mathbf{X}|] = \sum_{i=1}^{T+1} \frac{\lambda_i^\mathcal{X}}{\lambda_i^\mathcal{X} + 1} \tag{7}$$

where $\mathcal{X}$ is the set of states in $\tau$, $L(\mathcal{X})$ is the kernel matrix built with feature vectors of states in $\mathcal{X}$, $f(\tau)$ is the expected number of modes (landmark states) covered in a trajectory $\tau$. Thus, $\mathcal{L}_1^{DPP}$ measures single-option coverage, by maximizing which we can enhance exploration of each option.

As for the kernel matrix, according to Section 2.2, we can construct $L(\mathcal{X})$ by defining the quality measure $\overrightarrow{q}$ and normalized feature vector for each state in $\mathcal{X}$. States with higher expected returns should be visited more frequently and thus be assigned with higher quality values. However, given that there is no prior knowledge on the reward function, we assign equal quality measure to each state as 1. As for the features of each state, we define them using the Laplacian spectrum, i.e., eigenvectors corresponding to the $D$-smallest eigenvalues of the Laplacian matrix of the state transition graph, denoted as $[\overrightarrow{v_1}, \cdots, \overrightarrow{v_D}]$. To be specific, for each state $s$, its normalized feature is defined as: $\overrightarrow{b}(s) = [\overrightarrow{v_1}(s), \cdots, \overrightarrow{v_D}(s)] / \sqrt{\sum_{j=1}^{D} (\overrightarrow{v_j}(s))^2}$. The reasons for this feature design are as follows: **(1)** As shown in Spectral Clustering [49], states with high similarity in this feature embedding fall in

---

**Algorithm 1** Unsupervised Option Discovery based on DPP (ODPP)

---
1: Initialize the prior network $P_\omega$, policy network $\pi_\theta$, variational decoder $P_\phi$, and trajectory set $\overrightarrow{\tau}$
2: **for** *each training episode* **do**
3:    **for** $i = 1, 2, \ldots, N$ **do**
4:       Sample $s_0^i \sim \mu(\cdot)$ and $c \sim P_\omega(\cdot|s_0^i)$ or use the ones from the previous episode to collect trajectories subject to the same option and starting point
5:       Collect a trajectory $\tau_i = (s_0^i, a_0^i, \cdots, s_{T-1}^i, a_{T-1}^i, s_T^i)$, where $a_t^i \sim \pi_\theta(\cdot|s_t^i, c)$
6:       $\overrightarrow{\tau} \longleftarrow \overrightarrow{\tau} \cup \{\tau_i\}$
7:    **end for**
8:    Update $P_\omega, \pi_\theta$ with PPO and $P_\phi$ with SGD, based on $\overrightarrow{\tau}$ and Eq. (11)-(13); $\overrightarrow{\tau} \longleftarrow \{\}$
9: **end for**

---

the same cluster. Under a DPP with this feature embedding, the sampled states with distinct features should belong to different clusters, i.e., the landmarks. Then, by maximizing $\mathcal{L}_1^{DPP}$ (i.e., the expected number of landmarks covered in $\mathcal{X}$), the agent is encouraged to traverse multiple clusters within the state space. **(2)** With this feature design, our algorithm generalizes of the Laplacian-based option discovery [12]. In [12], they set a threshold to partition the state space into two parts – the set of states with higher values in the Fiedler vector (i.e., $\overrightarrow{v_2}$) than the threshold is used as the initiation set and the other states are used as the termination set, then the option policy is trained to connect states within these two areas. We note that, as a special case of our algorithm, when setting the feature dimension $D = 2$, we can get similar options with the Laplacian-based methods through maximizing $\mathcal{L}_1^{DPP}$. Given that the eigenvector corresponding to the smallest eigenvalue of a Laplacian matrix is $\overrightarrow{v_1} = \vec{1}$, states with diverse feature embeddings encoded by $[\overrightarrow{v_1}, \overrightarrow{v_2}]$ (i.e., $D = 2$) tend to differ in $\overrightarrow{v_2}$. By maximizing $\mathcal{L}_1^{DPP}$, the options are trained to visit the states that are as distinct in $\overrightarrow{v_2}$ as possible in a trajectory, which is similar with the ones learned in [12]. We empirically demonstrate this in Section 4. Note that the Laplacian spectrum $[\overrightarrow{v_1}, \cdots, \overrightarrow{v_D}]$ for infinite-scale state space can be learned as a neural network through representation learning [50] which is introduced in Appendix C.3. Thus, our algorithm can be applied to tasks with large-scale state spaces.

Next, we expect the set of sampled trajectories related to the same option $c$ and starting from the same state $s_0$, i.e., $\overrightarrow{\tau}_{(s_0,c)}$, to be consistent and thus hard to be distinguished by a DPP, which is important given the stochasticity of the policy output. This is realized by minimizing the expected mode number in each $\overrightarrow{\tau}_{(s_0,c)}$: ($s_0 \sim \mu(\cdot)$, $c \sim P_\omega(\cdot|s_0)$, $\overrightarrow{\tau}_{(s_0,c)} \sim P_\theta(\cdot|s_0, c)$)

$$\mathcal{L}_2^{DPP}(\omega, \theta) = \mathop{\mathbb{E}}_{s_0, c, \overrightarrow{\tau}_{(s_0,c)}} \left[ g(\overrightarrow{\tau}_{(s_0,c)}) \right], \; g(\overrightarrow{\tau}_{(s_0,c)}) = \mathop{\mathbb{E}}_{\mathbf{Y} \sim P_{L(\mathcal{Y})}} [|\mathbf{Y}|] = \sum_{i=1}^{M} \frac{\lambda_i^{\mathcal{Y}}}{\lambda_i^{\mathcal{Y}} + 1} \quad (8)$$

where $\mathcal{Y}$ is the set of $M$ trajectories related to $c$ starting at $s_0$ (i.e., $\{\tau_1, \cdots, \tau_M\} = \overrightarrow{\tau}_{(s_0,c)}$), $L(\mathcal{Y})$ is the kernel matrix built with the feature vectors of each trajectory, i.e., $\overrightarrow{b(\tau_i)}$. The feature vector of a trajectory can be obtained based on the ones of its landmark states $G_i$ through the Structured DPP framework [51]: $\overrightarrow{b(\tau_i)} = \sum_{s \in G_i} \overrightarrow{b(s)}$. ($\overrightarrow{b(s)}$ is defined above.) Alternatively, we can use the hidden layer output of the decoder $P_\phi(c|G, s_0)$, which embeds the information contained in $G$. This design is commonly adopted in DPP-related works [52, 53]. We will compare these two choices in Section 4. Notably, our kernel matrix design, i.e, $L(\mathcal{X})$ and $L(\mathcal{Y})$, is task-irrelevant and domain-knowledge-free.

Last, the set of sampled trajectories subject to different options should be diverse. This is achieved by maximizing its expected mode number (i.e., Eq. (9)). Here, $\mathcal{Z}$ is the union set of $\mathcal{Y}$ related to different options, i.e., $\cup_{c'} \overrightarrow{\tau}_{(s_0,c')}$, and $L(\mathcal{Z})$ is the kernel matrix corresponding to trajectories in $\mathcal{Z}$.

$$\mathcal{L}_3^{DPP}(\omega, \theta) = \mathop{\mathbb{E}}_{s_0, c, \overrightarrow{\tau}_{(s_0,c)}} \left[ h(\cup_{c'} \overrightarrow{\tau}_{(s_0,c')}) \right], \; h(\cup_{c'} \overrightarrow{\tau}_{(s_0,c')}) = \mathop{\mathbb{E}}_{\mathbf{Z} \sim P_{L(\mathcal{Z})}} [|\mathbf{Z}|] = \sum_{i=1}^{K} \frac{\lambda_i^{\mathcal{Z}}}{\lambda_i^{\mathcal{Z}} + 1} \quad (9)$$

### 3.3 Overall Algorithm Framework

The overall objective is to maximize Eq. (10). The hyperparameters $\alpha_{1:3} \geq 0$ (provided in Appendix C.6) are the weights for each DPP term and can be fine-tuned to enable a tradeoff between coverage and diversity of the learned options.

$$\mathcal{L}(\omega, \theta, \phi) = \mathcal{L}^{IB}(\omega, \theta, \phi) + \alpha_1 \mathcal{L}_1^{DPP}(\omega, \theta) - \alpha_2 \mathcal{L}_2^{DPP}(\omega, \theta) + \alpha_3 \mathcal{L}_3^{DPP}(\omega, \theta) \quad (10)$$

Based on Eq. (10), we can calculate the gradients with respect to $\omega, \theta, \phi$, i.e., the parameters of the prior network, policy network and variational decoder, and then apply corresponding algorithms for optimization. First, $\nabla_\phi \mathcal{L} = \nabla_\phi \mathcal{L}^{IB}$, so $P_\phi$ can be optimized as a standard likelihood maximization problem with SGD [54]. Next, regarding $\omega$ and $\theta$, we have Proposition 2, which is proved in Appendix A.3. Notably, we select PPO [55] to update $\omega$ and $\theta$. The overall algorithm (ODPP) is summarized as Algorithm 1, where the main learning outcome is the intra-option policy $\pi_\theta$. When applied to downstream tasks, $\pi_\theta(a|s,c)$ can be fixed and we only need to learn a high-level policy $P_\psi(c|s)$ to select among options. We note that our algorithm is highly salable and only slightly increases the time complexity compared with previous algorithms in this field. Detailed discussion on the complexity and scalability are provided in Appendix C.4 and C.5.

**Proposition 2.** *The gradients of the overall objective $\mathcal{L}(\omega, \theta, \phi)$ with respect to $\omega$ and $\theta$ can be unbiasedly estimated using Eq. (11). Here, $Q^{P_\omega}$ and $Q_m^{\pi_\theta}$ are the Q-functions for the prior and policy networks, respectively. Consequently, both networks can be trained using reinforcement learning.*

$$\nabla_\omega \mathcal{L} = \mathop{\mathbb{E}}_{s_0,c} \left[ \nabla_\omega \log P_\omega(c|s_0) Q^{P_\omega} \right], \; \nabla_\theta \mathcal{L} = \mathop{\mathbb{E}}_{s_0,c,\overrightarrow{\tau}} \left[ \sum_{m=1}^{M} \sum_{t=0}^{T-1} \nabla_\theta \log \pi_\theta(a_t^m | s_t^m, c) Q_m^{\pi_\theta} \right] \quad (11)$$

$$Q^{P_\omega}(c, s_0) = -\log P_\omega(c|s_0) + \mathop{\mathbb{E}}_{\overrightarrow{\tau}} \left[ \sum_{m=1}^{M} Q_m^{\pi_\theta}(\overrightarrow{\tau}, s_0, c) \right] \quad (12)$$

$$Q_m^{\pi_\theta}(\overrightarrow{\tau}, c, s_0) = \frac{P^{DPP}(G_m|\tau_m) \log P_\phi(c|s_0, G_m)}{M} - \frac{\beta}{M} \sum_{t=0}^{T-1} \log \pi_\theta(a_t^m|s_t^m, c)$$
$$+ \frac{\alpha_1}{M} f(\tau_m) - \alpha_2 g(\overrightarrow{\tau}_{(s_0,c)}) + \alpha_3 h(\bigcup_{c'} \overrightarrow{\tau}_{(s_0,c')}) \quad (13)$$

## 4 Evaluation and Main Results

In this section, we compare ODPP with SOTA baselines on a series of RL tasks. **(1)** For intuitive visualization, we test these algorithms on maze tasks built with Mujoco [56]. In particular, we select the Point and Ant as training agents, and put them in complex 3D Mujoco Maze environments (Figure 4(a) and 4(d)). We evaluate the diversity and coverage of the options learned with different algorithms, through visualizing corresponding trajectories. Then, we provide a quantitative study to see if these options can aid learning in downstream tasks – goal-achieving or exploration tasks in the maze environments. Both tasks are long-horizon with an episode length of 500. **(2)** To show the applicability of our algorithm on a wide range of tasks, we test it on 3D Ant locomotion tasks (Figure 5(a)) and Atari video games. In this part, we focus on evaluating if the agent can learn effective skills without supervision of task-specific rewards.

As mentioned in Section 2.1, our algorithm follows the option-first variational option discovery, which starts from a random policy rather than an efficient exploration policy like the trajectory-first methods. To keep it fair, we compare our algorithm with several SOTA option-first methods: VIC [7], DIAYN [8], VALOR [27], DADS [9], and APS [57]. Further, our algorithm integrates the variational and Laplacian-based option discovery, so we compare it with a SOTA Laplacian-based method as well: DCO [12]. The codes are available at https://github.com/LucasCJYSDL/ODPP.

### 4.1 Ablation Study

In Figure 2, we visualize trajectories of options learned with different algorithms in the Point Room task, which start from the same initial state and have the same horizon (i.e., 50 steps). Note that the visualizations in Figure 1-3 are aerial views of the 3D Mujoco Room/Corridor. As mentioned in Section 3.2, the objective of DCO is to train options that can connect states with high and low values in the Fielder vector of the state transition graph. Due to its algorithm design [12], we can only learn one option with DCO at a time. While, with the others, we can learn multiple options simultaneously. **(1)** From (a)-(c), we can see that variational option discovery methods can discover diverse options, but these options can hardly approach the "bottleneck" states in the environment which restricts their coverage of the state space. On the contrary, in (d), options trained with DCO can go through two "bottleneck" states but lack diversity, since they stick to the direction given by the Fielder vector. While, as shown in (h), options learnt with our algorithm have both superior diversity

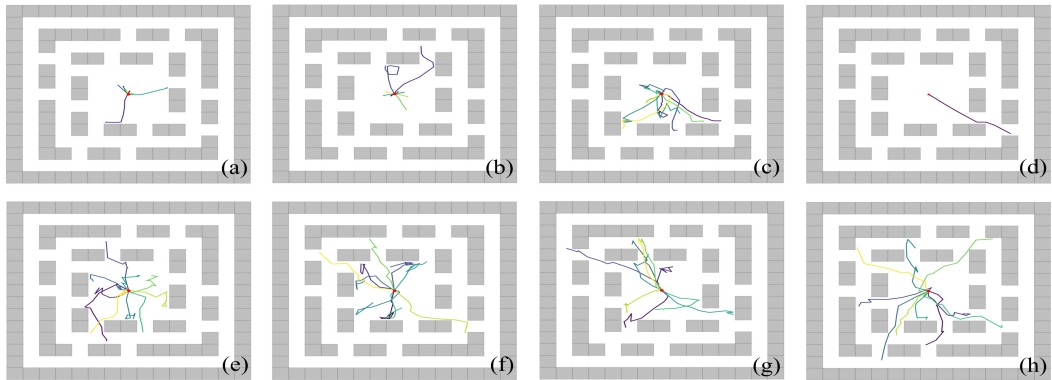

Figure 2: (a) VIC, (b) DIAYN, (c) VALOR, (d) DCO, (e) ODPP ($\mathcal{L}^{IB}$), (f) ODPP ($\mathcal{L}^{IB}, \mathcal{L}_1^{DPP}$), (g) ODPP using trajectory features defined by hidden layer output, (h) ODPP using trajectory features defined with the Structured DPP framework.

and coverage. **(2)** In (e), we can already get significant better options than the baselines by only using $\mathcal{L}^{IB}$ as the objective. From (e) to (f), it can be observed that additionally introducing the objective term $\mathcal{L}_1^{DPP}$ can encourage options to cover more landmark states. From (f) to (h), we further add $\mathcal{L}_2^{DPP}$ and $\mathcal{L}_3^{DPP}$, which makes the option trajectories more distinguishable from the view of DPP. Also, in (g) and (h), we adopt different designs of the trajectory feature. It shows that using trajectory features defined with the Structured DPP framework (introduced in Section 3.2) is better than using the hidden layer output of the decoder $P_\phi$. These results show the effectiveness of each component in our algorithm design and demonstrate that our algorithm can construct options with higher diversity and coverage than baselines. In Appendix D.3, we provide quantitative results to further show the improvement brought by each objective term of ODPP.

Next, as claimed in Section 3.2, if setting the feature dimension $D = 2$, ODPP is expected to learn similar options with DCO, thus ODPP generalizes the Laplacian-based method. In Figure 3, we visualize the value of each state in the Fiedler vector as the background color (the darker the higher), and the options learned with DCO and ODPP in the Point Corridor task starting from different locations. We can observe from (a) and (b) that most of the learned options are similar both in direction and

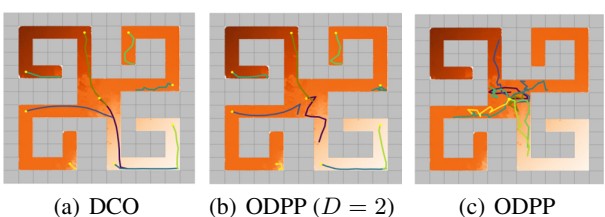

(a) DCO      (b) ODPP ($D=2$)      (c) ODPP

Figure 3: (a) Options learned with DCO; (b) Options learned with ODPP, when setting the feature dimension as 2; (c) Options learned with ODPP in the normal setting.

length. Further, if we adopt the normal setting, where $D = 30$ and the number of options to learn at a time is 10, we can get diverse options shown as (c), which can be beneficial for various downstream tasks. In this case, ODPP can be viewed as a generalization and extension of DCO through variational tools (e.g., $\mathcal{L}^{IB}$) to learn multiple diverse options at a time.

## 4.2 Evaluation in Downstream Tasks

In Figure 4, we evaluate the options learned with different algorithms on a series of downstream tasks. These options are trained without task-specific rewards, and thus potentially applicable to different downstream tasks in the environment. Firstly, we test the options in Point Room/Corridor goal-achieving tasks where a point agent is trained to achieve a certain goal (i.e., red points in Figure 4(a) and 4(d)). This task is quite challenging since: (1) The location of the goal is not included in the observation. (2) The agent will get a positive reward only when it reaches the goal area; otherwise, it will receive a penalty. Hence, the reward setting is highly sparse and delayed, and the agent needs to fully explore the environment for the goal state. In Figure 4(b)-4(c), we compare the mean and standard deviation of the performance (i.e., mean return of a decision step) of different algorithms

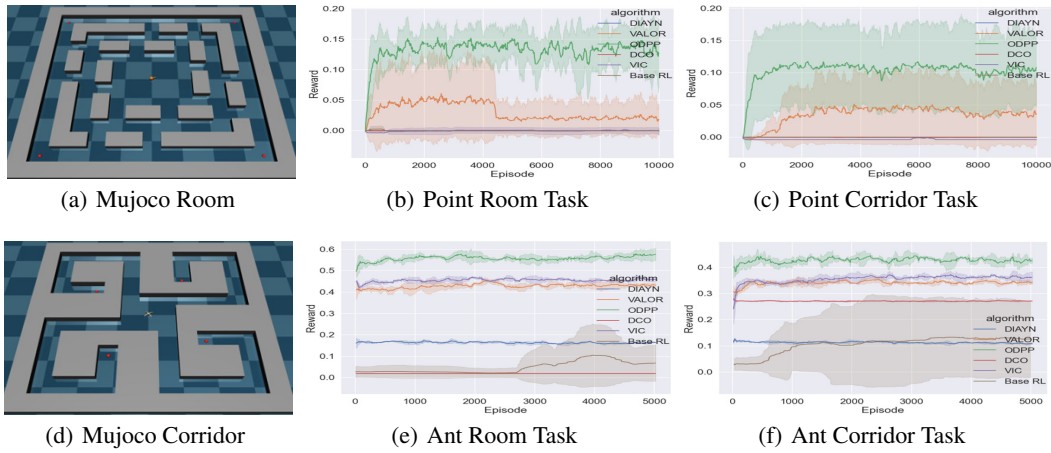

| (a) Mujoco Room | (b) Point Room Task | (c) Point Corridor Task |
| (d) Mujoco Corridor | (e) Ant Room Task | (f) Ant Corridor Task |

Figure 4: (a)(d) Mujoco Maze tasks. (b)(c) Applying the options to goal-achieving tasks in the Point Room/Corridor where the agent needs to achieve one of the four goals (red points). (e)(f) Applying options to exploration tasks in the Ant Room/Corridor where the agent needs to explore as far as possible to get higher reward.

in the training process, which is repeated four times (each time with a different goal). It can be observed that, with options learned by ODPP, the convergence speed and return value can be much higher. Note that the pretrained options are fixed for downstream tasks and we only need to learn an option selector $P_\psi(c|s)$ which gives out option choice $c$ at state $s$. In this way, we can simplify a continuous control task to a discrete task, and the advantage can be shown through the comparison with using PPO (i.e., "Base RL") directly on the task. To keep it fair, the PPO agent is pretrained for the same number of episodes as the option learning. Moreover, we evaluate these algorithms in Ant Room/Corridor exploration tasks where an Ant agent is trained to explore the areas as far from the start point (center) as possible. The reward for a trajectory is defined with the largest distance that the agent has ever reached during this training episode. In Figure 4(e)-4(f), we present the change of the trajectory return during the training process of these algorithms (repeated five times with different random seeds). The options trained with ODPP provide a good initialization of the policy for this exploration task. With this policy, the Ant agent can explore a much larger area in the state space.

As mentioned in Section 3, we learn a prior network $P_\omega$ together with the option policy network $\pi_\theta$. This design is different from previous algorithms which choose to fix the prior distribution. In Appendix D.1, we provide a detailed discussion on this and empirically show that we can get a further performance improvement in the downstream task by initializing the option selector $P_\psi$ with $P_\omega$.

### 4.3 Performance of Unsupervised Skill Discovery Across Various Benchmarks

In the 3D Locomotion task, an Ant agent needs to coordinate its four legs to move. In Figure 5(a), we visualize two of the controlling behaviors, learned by ODPP without supervision of any extrinsic rewards. Please refer to Appendix D.2 for more visualizations. The picture above shows that the Ant rolls to the right by running on Leg 1 first and then Leg 4, which is a more speedy way to move ahead. While, the picture below shows that it learns how to walk to the right by stepping on its front legs (2&3) and back legs (1&4) by turn. These complex behaviors are learned with only intrinsic objectives on diversity and coverage, which is quite meaningful. Further, we show numeric results to evaluate the learned behaviors. We use the reward defined in [58] as the metric, which is designed to encourage the Ant agent to move as fast as possible at the least control cost. In Figure 5(b), we take the average of five different random seeds, and the result shows that options trained by ODPP outperform the baselines. The reward drops during training for some baselines, which is reasonable since this is not the reward function used for option learning. Finally, to see whether we can learn a large number of options in the meantime with ODPP, we test the performance of the discovered options when setting the number of options to learn as 10, 20, 40, 60. From Figure 5(c), it can be observed that even when learning a large number of options at the same time, we can still get options with high quality (mean) and diversity (standard deviation) which increase during the training.

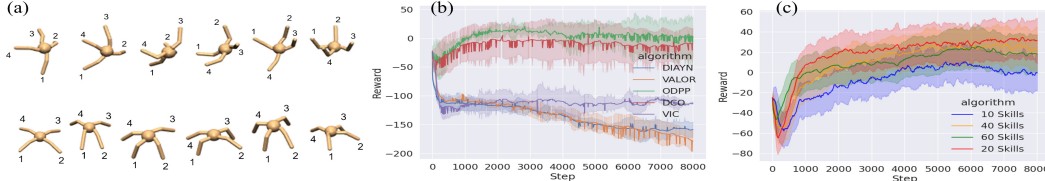

Figure 5: (a) Visualization of the controlling behaviors learned by ODPP. (b) The change of the mean and standard deviation of the trajectory rewards subject to different options in the training process. (c) The performance change as the number of options to learn at the same time goes up.

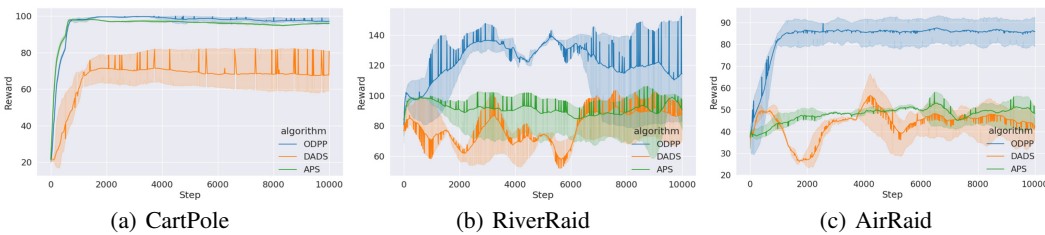

     (a) CartPole                       (b) RiverRaid                     (c) AirRaid

Figure 6: Evaluation results on OpenAI Gym and Atari games to show the superiority of our algorithm on more general tasks. Our algorithm performs the best in all the three tasks, and the performance improvement becomes more significant as the task difficulty increases.

To further demonstrate the applicability of ODPP, we compare it with SOTA baselines on more general OpenAI Gym and Atari tasks [59]. Notably, we adopt two more advanced skill discovery algorithms as baselines: DADS and APS. Comparisons among these two and previous baselines are in Appendix D.4. Skills discovered with different algorithms are evaluated with reward functions carefully-crafted for each task, provided by OpenAI Gym. For each algorithm, we learn 10 skills, of which the average cumulative rewards (i.e., sum of rewards within a skill duration) in the training process are shown in Figure 6. The skill duration is set as 100 for CartPole and 50 for the other two. Note that the complete episode horizon is 200 for CartPole and 10000 for AirRaid and RiverRaid. Thus, it would be unfair to compare the cumulative reward of a skill with the one of a whole episode. Our algorithm performs the best in all the three tasks, and the improvement becomes more significant as the task difficulty increases. When relying solely on MI-based objectives, the agent tends to reinforce already discovered behaviors for improved diversity rather than exploration. The explicit incorporation of coverage and diversity objectives in our algorithm proves beneficial in this case.

## 5 Conclusion and Discussion

ODPP is a novel unsupervised option discovery framework based on DPP, which unifies variational and Laplacian-based option discovery methods. Building upon the information-theoretic objectives in prior variational research, we propose three DPP-related measures to explicitly quantify and optimize diversity and coverage of the discovered options. Through a novel design of the DPP kernel matrix based on the Laplacian spectrum of the state transition graph, ODPP generalizes SOTA Laplacian-based option discovery algorithms. We demonstrate the superior performance of ODPP over SOTA baselines using variational and Laplacian-based methods on a series of challenging benchmarks. Regarding limitations of ODPP, the primary one is to assign suitable weights for each objective term: $\alpha_{1:3}$ in Eq. (10) and $\beta$ in Eq. (6). These hyperparameters are crucial, yet conducting a grid search for their joint selection would be exhaustive. Instead, we employ a sequential, greedy method to select hyperparameters in accordance with our ablation study's procedure, further detailed in Appendix C.6.

## Acknowledgments and Disclosure of Funding

This project is supported by ONR award N00014-23-1-2850 and CISCO research award 76934189.

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

# A  Notations and Proof

## A.1  Basic Concepts and Notations

**Markov Decision Process (MDP):** The reinforcement learning problem can be described with an MDP, denoted by $\mathcal{M} = (\mathcal{S}, \mathcal{A}, \mathcal{P}, \mathcal{R}, \gamma)$, where $\mathcal{S}$ is the state space, $\mathcal{A}$ is the action space, $\mathcal{P} : \mathcal{S} \times \mathcal{A} \times \mathcal{S} \to [0, 1]$ is the state transition function, $\mathcal{R} : \mathcal{S} \times \mathcal{A} \to R^1$ is the reward function, and $\gamma \in (0, 1]$ is the discount factor.

**State transition graph in an MDP:** The state transitions in $\mathcal{M}$ can be modelled as a state transition graph $G = (V_G, E_G)$, where $V_G$ is a set of vertices representing the states in $\mathcal{S}$, and $E_G$ is a set of undirected edges representing state adjacency in $\mathcal{M}$. We note that:

**Remark.** *There is an edge between state $s$ and $s'$ (i.e., $s$ and $s'$ are adjacent) if and only if $\exists\, a \in \mathcal{A},\ s.t.\ \mathcal{P}(s, a, s') > 0 \ \vee\ \mathcal{P}(s', a, s) > 0$.*

The adjacency matrix $A$ of $G$ is an $|\mathcal{S}| \times |\mathcal{S}|$ matrix whose $(i, j)$ entry is 1 when $s_i$ and $s_j$ are adjacent, and 0 otherwise. The degree matrix $D$ is a diagonal matrix whose entry $(i, i)$ equals the number of edges incident to $s_i$. The Laplacian matrix of $G$ is defined as $L = D - A$. Its second smallest eigenvalue $\lambda_2(L)$ is called the algebraic connectivity of the graph $G$, and the corresponding normalized eigenvector is called the Fiedler vector [60]. Last, the normalized Laplacian matrix is defined as $\mathcal{L} = D^{-\frac{1}{2}} L D^{-\frac{1}{2}}$.

## A.2  Proof of Proposition 1

To start with, we can find a lower bound of the second term in Eq. (5) as follows:

$$
\begin{aligned}
-\beta \mathop{\mathbb{E}}_{s_0 \sim \mu(\cdot)} [I(c, \tau | s_0)] &= -\beta \mathop{\mathbb{E}}_{s_0 \sim \mu(\cdot)} \left[ -\sum_\tau P(\tau|s_0) \log P(\tau|s_0) + \sum_{c,\tau} P(c, \tau|s_0) \log P_\theta(\tau|s_0, c) \right] \\
&= -\beta \mathop{\mathbb{E}}_{s_0 \sim \mu(\cdot)} \left[ -\sum_{c,\tau} P(c, \tau|s_0) \log P(\tau|s_0) + \sum_{c,\tau} P(c, \tau|s_0) \log P_\theta(\tau|s_0, c) \right] \\
&= -\beta \mathop{\mathbb{E}}_{s_0 \sim \mu(\cdot)} \left[ \sum_{c,\tau} P_\omega(c|s_0) P_\theta(\tau|c, s_0) \log \frac{P_\theta(\tau|s_0, c)}{P(\tau|s_0)} \right] \\
&= -\beta \mathop{\mathbb{E}}_{s_0 \sim \mu(\cdot)} \left[ \sum_{c,\tau} P_\omega(c|s_0) P_\theta(\tau|c, s_0) \left[ \log \frac{P_\theta(\tau|s_0, c)}{Unif(\tau|s_0)} - \log \frac{P(\tau|s_0)}{Unif(\tau|s_0)} \right] \right] \\
&= -\beta \mathop{\mathbb{E}}_{s_0 \sim \mu(\cdot)} \left[ \left[ \sum_{c,\tau} P_\omega(c|s_0) P_\theta(\tau|c, s_0) \log \frac{P_\theta(\tau|s_0, c)}{Unif(\tau|s_0)} \right] - D_{KL}(P(\tau|s_0) \| Unif(\tau|s_0)) \right] \\
&\geq -\beta \mathop{\mathbb{E}}_{s_0 \sim \mu(\cdot)} \left[ \sum_{c,\tau} P_\omega(c|s_0) P_\theta(\tau|c, s_0) \log \frac{P_\theta(\tau|s_0, c)}{Unif(\tau|s_0)} \right] \\
&= -\beta \mathop{\mathbb{E}}_{\substack{s_0 \sim \mu(\cdot) \\ c \sim P_\omega(\cdot|s_0)}} [D_{KL}(P_\theta(\tau|s_0, c) \| Unif(\tau|s_0))]
\end{aligned}
\tag{14}
$$

where $D_{KL}(\cdot)$ denotes the Kullback-Leibler (KL) Divergence which is non-negative, $P_\theta(\tau|s_0, c) = \prod_{t=0}^{T-1} \pi_\theta(a_t|s_t, c) P(s_{t+1}|s_t, a_t)$ is the probability of the trajectory $\tau$ given $s_0$ and $c$ under the option policy $\pi_\theta$, and $Unif(\tau|s_0)$ is the probability of the same trajectory given $s_0$ under the random walk policy. Instead of explicitly calculating $P(\tau|s_0)$ which is impractical, we introduce $Unif(\tau|s_0)$ to convert the second term in Eq. (5) into a regularization term to encourage exploration and diversity.

As for the first term in Eq. (5), we can deal with it as Eq. (15), where we introduce $P_\phi(c|s_0, G)$ as the variational estimation of $P(c|s_0, G)$ which is hard to acquire. The first inequality in Eq. (15) is based on the fact that KL Divergence is non-negative. While, the second inequality holds because we only keep the trajectory $\tau$ from which $G$ is sampled, so the trajectory and its corresponding landmark states form a bijection.

$$\mathbb{E}_{s_0 \sim \mu(\cdot)} [I(c, G|s_0)] = \mathbb{E}_{s_0 \sim \mu(\cdot)} \left[ -\sum_c P_\omega(c|s_0) \log P_\omega(c|s_0) + \sum_{c,G} P_\omega(c|s_0) P_\theta(G|s_0, c) \log P(c|s_0, G) \right]$$

$$= \mathbb{E}_{s_0 \sim \mu(\cdot)} \left[ -\sum_c P_\omega(c|s_0) \log P_\omega(c|s_0) + \sum_{c,G} P_\omega(c|s_0) P_\theta(G|s_0, c) \log \left[ P_\phi(c|s_0, G) \frac{P(c|s_0, G)}{P_\phi(c|s_0, G)} \right] \right]$$

$$= \mathcal{H}(\mathcal{C}|\mathcal{S}) + \mathbb{E}_{s_0 \sim \mu(\cdot)} \left[ \sum_{c,G} P_\omega(c|s_0) P_\theta(G|s_0, c) \log P_\phi(c|s_0, G) \right] + \sum_G P(G|s_0) D_{KL}(P(c|s_0, G) || P_\phi(c|s_0, G))$$

$$\geq \mathcal{H}(\mathcal{C}|\mathcal{S}) + \mathbb{E}_{s_0 \sim \mu(\cdot)} \left[ \sum_{c,G} P_\omega(c|s_0) P_\theta(G|s_0, c) \log P_\phi(c|s_0, G) \right]$$

$$= \mathcal{H}(\mathcal{C}|\mathcal{S}) + \mathbb{E}_{s_0 \sim \mu(\cdot)} \left[ \sum_{c,G} P_\omega(c|s_0) \left[ \sum_{\tau'} P_\theta(\tau'|s_0, c) P^{DPP}(G|\tau') \right] \log P_\phi(c|s_0, G) \right]$$

$$\geq \mathcal{H}(\mathcal{C}|\mathcal{S}) + \mathbb{E}_{s_0 \sim \mu(\cdot)} \left[ \sum_{c,\tau} P_\omega(c|s_0) P_\theta(\tau|s_0, c) P^{DPP}(G|\tau) \log P_\phi(c|s_0, G) \right]$$

$$\tag{15}$$

### A.3 Proof of Proposition 2

First, we take the gradient with respect to $\omega$ and get the following result:

$$\nabla_\omega \mathcal{L} = \mathbb{E}_{s_0 \sim \mu(\cdot)} \Big[ -\sum_c [\nabla_\omega P_\omega(c|s_0) \log P_\omega(c|s_0) + \nabla_\omega P_\omega(c|s_0)]$$
$$+ \sum_{c,\tau} \nabla_\omega P_\omega(c|s_0) P_\theta(\tau|s_0, c) P^{DPP}(G|\tau) \log P_\phi(c|s_0, G)$$
$$- \beta \sum_c \nabla_\omega P_\omega(c|s_0) D_{KL}(P_\theta(\tau|s_0, c) || Unif(\tau|s_0))$$
$$+ \alpha_1 \sum_{c,\tau} \nabla_\omega P_\omega(c|s_0) P_\theta(\tau|s_0, c) f(\tau)$$
$$- \alpha_2 \sum_c \nabla_\omega P_\omega(c|s_0) \sum_{\overrightarrow{\tau}(s_0, c)} P_\theta(\overrightarrow{\tau}|s_0, c) g(\overrightarrow{\tau}(s_0, c))$$
$$+ \alpha_3 \sum_c \nabla_\omega P_\omega(c|s_0) \sum_{\overrightarrow{\tau}(s_0, c)} P_\theta(\overrightarrow{\tau}|s_0, c) h(\cup_{c'} \overrightarrow{\tau}(s_0, c'))\Big]$$

$$\tag{16}$$

Given that $\nabla_\omega P_\omega(c|s_0) = P_\omega(c|s_0) \nabla_\omega \log P_\omega(c|s_0)$ and the definition of KL Divergence, i.e., $D_{KL}(P_\theta(\tau|s_0, c) || Unif(\tau|s_0)) = \sum_\tau P_\theta(\tau|s_0, c) \sum_{t=0}^{T-1} [\log \pi_\theta(a_t|s_t, c) - \log \pi_{unif}(a_t|s_t)]$, we can simplify Eq. (16) as:

$$\nabla_\omega \mathcal{L} = \mathbb{E}_{\substack{s_0 \sim \mu(\cdot) \\ c \sim P_\omega(\cdot|s_0)}} \left[ \nabla_\omega \log P_\omega(c|s_0) Q^{P_\omega}(c, s_0) \right] \tag{17}$$

where the related Q-function $Q^{P_\omega}(c, s_0)$ is defined as:

$$Q^{P_\omega}(c, s_0) = -\log P_\omega(c|s_0) + \mathbb{E}_{\overrightarrow{\tau}(s_0, c) \sim P_\theta(\cdot|s_0, c)} \left[ -\alpha_2 g(\overrightarrow{\tau}(s_0, c)) + \alpha_3 h(\cup_{c'} \overrightarrow{\tau}(s_0, c')) \right]$$
$$+ \mathbb{E}_{\tau \sim P_\theta(\cdot|s_0, c)} \left[ P^{DPP}(G|\tau) \log P_\phi(c|s_0, G) - \beta \sum_{t=0}^{T-1} \log \pi_\theta(a_t|s_t, c) + \alpha_1 f(\tau) \right]$$

$$\tag{18}$$

Next, we calculate the gradient with respect to $\theta$ as follows:

$$\nabla_\theta \mathcal{L} = \underset{s_0 \sim \mu(\cdot)}{\mathbb{E}} \sum_{c,\tau} P_\omega(c|s_0) \nabla_\theta P_\theta(\tau|s_0,c) P^{DPP}(G|\tau) \log P_\phi(c|s_0,G)$$

$$- \beta \sum_{c,\tau} P_\omega(c|s_0) \nabla_\theta P_\theta(\tau|s_0,c) \sum_{t=0}^{T-1} [\log \pi_\theta(a_t|s_t,c) - \log \pi_{unif}(a_t|s_t)]$$

$$- \beta \sum_{c,\tau} P_\omega(c|s_0) P_\theta(\tau|s_0,c) \sum_{t=0}^{T-1} \nabla_\theta \log \pi_\theta(a_t|s_t,c)$$

$$+ \alpha_1 \sum_{c,\tau} P_\omega(c|s_0) \nabla_\theta P_\theta(\tau|s_0,c) f(\tau)$$

$$+ \sum_c P_\omega(c|s_0) \sum_{\overrightarrow{\tau}(s_0,c)} \nabla_\theta P_\theta(\overrightarrow{\tau}|s_0,c) \left[-\alpha_2 g(\overrightarrow{\tau}(s_0,c)) + \alpha_3 h(\underset{c'}{\cup}\overrightarrow{\tau}(s_0,c'))\right]$$

(19)

With $\nabla_\theta P_\theta(\tau|s_0,c) = P_\theta(\tau|s_0,c)\nabla_\theta \log P_\theta(\tau|s_0,c) = P_\theta(\tau|s_0,c)\sum_{t=0}^{T-1}\nabla_\theta \log \pi_\theta(a_t|s_t,c)$, and $\nabla_\theta P_\theta(\overrightarrow{\tau}|s_0,c) = P_\theta(\overrightarrow{\tau}|s_0,c)\sum_{m=1}^{M}\sum_{t=0}^{T-1}\nabla_\theta \log \pi_\theta(a_t^m|s_t^m,c)$ where $s_t^m$ ($a_t^m$) is the state (action) at step $t$ in trajectory $m$, Eq. (19) can be written as follows:

$$\nabla_\theta \mathcal{L} = \underset{s_0,c,\tau}{\mathbb{E}} \left[\sum_{t=0}^{T-1} \nabla_\theta \log \pi_\theta(a_t|s_t,c)\left[P^{DPP}(G|\tau)\log P_\phi(c|s_0,G) - \beta\sum_{t=0}^{T-1}\log \pi_\theta(a_t|s_t,c) + \alpha_1 f(\tau)\right]\right]$$

$$+ \underset{s_0,c,\overrightarrow{\tau}}{\mathbb{E}} \left[\sum_{m=1}^{M}\sum_{t=0}^{T-1} \nabla_\theta \log \pi_\theta(a_t^m|s_t^m,c)\left[-\alpha_2 g(\overrightarrow{\tau}_{(s_0,c)}) + \alpha_3 h(\underset{c'}{\cup}\overrightarrow{\tau}_{(s_0,c')})\right]\right]$$

$$= \underset{s_0,c,\overrightarrow{\tau}}{\mathbb{E}} \left[\sum_{m=1}^{M}\sum_{t=0}^{T-1} \nabla_\theta \log \pi_\theta(a_t^m|s_t^m,c) Q_m^{\pi_\theta}(\overrightarrow{\tau},s_0,c)\right]$$

(20)

where the advantage term is as Eq. (21), $\overrightarrow{\tau} = \{\tau_1,\cdots,\tau_M\}$, $\tau_m = (s_0^m, a_0^m, \cdots, s_{T-1}^m, a_{T-1}^m, s_T^m)$:

$$Q_m^{\pi_\theta}(\overrightarrow{\tau},s_0,c) = \frac{P^{DPP}(G_m|\tau_m)\log P_\phi(c|s_0,G_m)}{M} - \frac{\beta}{M}\sum_{t=0}^{T-1}\log \pi_\theta(a_t^m|s_t^m,c)$$

$$+ \frac{\alpha_1}{M}f(\tau_m) - \alpha_2 g(\overrightarrow{\tau}_{(s_0,c)}) + \alpha_3 h(\underset{c'}{\cup}\overrightarrow{\tau}_{(s_0,c')})$$

(21)

Then, it's not hard to see the relationship between $Q^{P_\omega}$ and $Q_m^{\pi_\theta}$ as:

$$Q^{P_\omega}(c,s_0) = -\log P_\omega(c|s_0) + \underset{\overrightarrow{\tau}}{\mathbb{E}}\left[\sum_{m=1}^{M} Q_m^{\pi_\theta}(\overrightarrow{\tau},s_0,c)\right]$$

(22)

# B    Comparisons with Recent Variational Option Discovery Algorithms

Here, we provide comparisons of our algorithm with more recent variational option discovery methods [30, 31, 32, 33, 34, 35].

In [30], the authors focus on alternative approaches to leverage options learned during the unsupervised phase, rather than new option discovery algorithms. They still adopt objectives based on Mutual Information (MI) as previous works without solving the exploration issue.

The authors of [31] propose a slightly-modified version of VIC to improve usefulness of the discovered options by introducing an extra posterior. Still, their options are not explicitly trained for better coverage/exploration like ours.

The authors of [32] propose to replace the fixed prior distribution $P(c)$ with a fixed dynamics model over the option latent codes $P(c_t|c_{t-1})$. Each latent code corresponds to a sub-trajectory. By concatenating sub-trajectories, the agent can reach much further states. They only rely on MI-based objectives, so they cannot model the coverage of each option (like ours) and instead choose to chain options for better overall coverage. The fixed $P(c_t|c_{t-1})$ can result in inflexibility when applying options in downstream tasks.

In [33], they employ a multi-step protocol to generate options organized in a tree-like structure. Heuristics and structural limits are involved in each step, which may hinder its generality. Also, they propose to optimize the local coverage around the final state rather than the overall trajectory coverage like ours.

The authors of [34] note that, with variational objectives proposed in VIC and DIAYN, the agent can be discouraged from seeking out new states, since the variational posterior $P_\phi(c|s)$ is likely to make poor predictions when presented with trajectories containing previously unseen states, resulting in low rewards for the policy. Thus, an exploration bonus should be introduced as an extra reward term. In [34], they choose to train ensembles of discriminators and adopt their disagreement as a measure of state uncertainty. States of higher uncertainty are assigned with higher exploration bonus. Similar to theirs, our algorithm introduces additional bonus as a reimbursement of the unnecessary pessimistic exploration. The difference lies that they adopt an implicit exploration measure (i.e., the disagreement among ensembles) while we use explicit ones based on DPP.

In [35], they optimize the MI $I(s,c) = \mathcal{H}(s) - \mathcal{H}(s|c)$, where $\mathcal{H}(s)$ is for improving exploration of the learned options. They adopt a variational posterior $P_\phi(s|c)$ to estimate $\mathcal{H}(s|c)$. Learning $P_\phi(s|c)$ can be challenging when the state space is high-dimensional, compared with $P_\phi(c|s)$ used in our paper, which can hinder the optimization. Further, in DADS, they categorize algorithms utilizing $I(s,c) = \mathcal{H}(s) - \mathcal{H}(s|c)$ as the forward form of MI-based option discovery, and they empirically and theoretically show the limited capability of these algorithms for exploration even with $\mathcal{H}(s)$ in the objective.

# C    Implementation Details and Analysis of ODPP

## C.1    The Choice of Diversity Measure

The expected cardinality is a better choice for the diversity measure than the likelihood shown as Eq. (1). Using the log-likelihood based on the Determinant of the DPP kernel matrix directly would heavily penalize repeated items in the sampled set $\mathcal{W}$ in Eq. (1). For example, if there are very similar points in $\mathcal{W}$, the corresponding rows in the kernel matrix will be almost identical and lead to a zero determinant, which will cause numerical issues for the logarithm function. Take our work as an example: at the beginning of the training stage, the moving range of the Mujoco agent is very limited, then we always include very close states in a trajectory, which will always lead to a zero determinant and thus cannot provide training signals. However, the expected cardinality only counts the number of diverse states in a trajectory that will not be heavily influenced by the repeated items. Therefore, we select the expected cardinality as the diversity measure in this paper.

## C.2    Fast Greedy MAP Inference for DPP

The maximum a posteriori (MAP) inference of DPP aims at finding the subset of items with the highest possibility under the DPP measure, which is NP-hard [48]. The log-probability function in

DPP, i.e., $l(W) = \log \det(L_W)$, is submodular, which means:

$$\forall\, i \in \mathcal{W},\ W_1 \subseteq W_2 \subseteq \mathcal{W}\backslash\{i\},\ l(W_1 \cup \{i\}) - l(W_1) \geq l(W_2 \cup \{i\}) - l(W_2) \qquad (23)$$

Thus, the MAP inference for DPP can be converted to a submodular maximization problem, where greedy algorithms have shown promising empirical success. Recently, the authors of [39] propose a fast greedy method for MAP inference in DPP with time complexity $\mathcal{O}(S^2 N)$ to return $S$ items out of a sample space of size $N$. The key step of their algorithm is that for each iteration, the item which maximizes the marginal gain:

$$j = \underset{i \in \mathcal{W}\backslash W_{map}}{\arg\max}\ l(W_{map} \cup \{i\}) - l(W_{map}) \qquad (24)$$

is added to $W_{map}$ starting from an initial set $\emptyset$, until the maximal marginal gain becomes negative or the target sample number is reached (i.e., *stopping criteria*). This part is not our contribution. We provide its detailed pseudo code as Algorithm 2. For the derivation, please refer to the original paper [39]. We also provide its implementation code as a part of the complete code of ODPP.

---

**Algorithm 2** Fast Greedy MAP Inference for DPP

---

1: **Input:** The set of items $\mathcal{W}$ and its kernel matrix $\mathbf{L}$, *stopping criteria*
2: **Initialize:** For $i \in \mathcal{W}$, $\mathbf{c}_i = []$, $d_i^2 = \mathbf{L}_{ii}$; $W_{map} = \{j\}$, where $j = \arg\max_{i \in \mathcal{W}} \log(d_i^2)$
3: **while** *stopping criteria* not satisfied **do**
4:    **for** $i \in \mathcal{W}\backslash W_{map}$ **do**
5:       $e_i = (\mathbf{L}_{ji} - <\mathbf{c}_j, \mathbf{c}_i>)/d_j$
6:       $\mathbf{c}_i = [\mathbf{c}_i\ e_i]$, $d_i^2 = d_i^2 - e_i^2$
7:    **end for**
8:    $j = \arg\max_{i \in \mathcal{W}\backslash W_{map}} \log(d_i^2)$, $W_{map} = W_{map} \cup \{j\}$
9: **end while**
10: **Return** $W_{map}$

---

### C.3   Computation of the Laplacian Spectrum for the Infinite-scale State Spaces

As mentioned in Section 3.2, the feature vector of each state $\overrightarrow{b_i}$ is defined with the eigenvectors corresponding to the $D$-smallest eigenvalues of the Laplacian matrix of the state transition graph. However, for the infinite-scale state spaces, we cannot obtain this Laplacian spectrum through matrix-based methods, so we adopt the NN-based method proposed in [50] for estimating the Laplacian spectrum, which has been proved to be scalable for infinite-scale state spaces and sufficiently accurate compared with the groundtruth. Since this algorithm is not our contribution, we only provide the take-away messages here for implementation.

According to [50], the $k$ smallest eigenvalues $\lambda_{1:k}$ and corresponding eigenvectors $v_{1:k}$ of the Laplacian $L$ can be estimated by: ($k = D$ for our case)

$$\min_{v_1, \cdots, v_k} \sum_{i=1}^{k} (k - i + 1) v_i^T L v_i,\ s.t.\ v_i^T v_j = \delta_{ij}, \forall\, i, j = 1, \cdots, k \qquad (25)$$

For the large-scale state space, the eigenvectors can be represented as a neural network that takes a state $s$ as input and outputs a $k$-dimension vector $[f_1(s), \cdots, f_k(s)]$ as an estimation of $[v_1(s), \cdots, v_k(s)]$. Accordingly, the objective in Equation (25) can be expressed as: (please refer to [50] for details)

$$G(f_1, \cdots, f_k) = \frac{1}{2}\mathbb{E}_{(s,s')\sim\mathcal{T}} \left[ \sum_{l=1}^{k}\sum_{i=1}^{l} (f_i(s) - f_i(s'))^2 \right] \qquad (26)$$

where $\mathcal{T}$ is a set of state-transitions collected by interacting with the environment through a random policy. Further, the orthonormal constraints in Equation (25) are implemented as a penalty term:

$$P(f_1, \cdots, f_k) = \alpha \mathbb{E}_{s\sim\rho, s'\sim\rho} \left[ \sum_{l=1}^{k}\sum_{i=1}^{l}\sum_{j=1}^{l} (f_i(s)f_j(s) - \delta_{ij})(f_i(s')f_j(s') - \delta_{ij}) \right] \qquad (27)$$

where $\alpha$ is the weight term and $\rho$ is the distribution of states in $\mathcal{T}$. To sum up, the eigenfunctions $f$ can be trained as an NN by minimizing the loss function:

$$L(f_1, \cdots, f_k) = G(f_1, \cdots, f_k) + P(f_1, \cdots, f_k) \qquad (28)$$

## C.4 Analysis on Computation Complexity

The learning target of ODPP is an intra-option policy $\pi_\theta(a|s, c)$ conditioned on the option choice $c$. As in Section 3.3, this policy is learned with an Actor-Critic algorithm for which the Q-function is defined as Eq. (13). This Q-function contains variational and DPP-based objectives. Compared with previous variational option discovery algorithms (e.g., DIAYN, VIC, VALOR), we additionally need to (a) sample landmark states from each trajectory and (b) calculate the DPP-related terms: $f(\cdot)$, $g(\cdot)$, $h(\cdot)$.

For (a), we adopt a fast greedy MAP inference algorithm for DPP. As mentioned in Appendix C.2, it takes $\mathcal{O}(S^2N)$ to sample $S$ landmark states from an option trajectory of length $N$. In our setting, $N = 50$, $S = 10$, so the process can be done in real-time.

For (b), we need to build DPP kernel matrices, and then compute $f(\cdot)$, $g(\cdot)$, $h(\cdot)$ based on eigenvalues of the corresponding kernel matrix as in Eq. (7)-(9). The time complexity for eigen decomposition is $\mathcal{O}(N^3)$, where $N$ is the size of the matrix. For the state kernel matrix, $N$ is the number of states in an option trajectory (i.e., the option horizon) which we set as 50. For the trajectory kernel matrix, $N$ corresponds to the number of trajectories collected in each training iteration, which is set as 100. Thus, $f(\cdot)$, $g(\cdot)$, $h(\cdot)$ can be computed in real-time.

To build the kernel matrix, we need feature vectors for each state. As introduced in Appendix C.3, the feature vector is the output of a pre-trained neural network which takes the state as input. The training of this feature function is based on state transitions in the replay buffer and only needs to be done for once or twice in the whole option discovery process, of which the time cost is within 30 minutes.

To sum up, compared with previous variational methods, we additionally introduce three DPP items to explicitly model the option diversity and coverage, of which the involvement only slightly increases the time complexity.

## C.5 Analysis on Scalability

ODPP can be adapted to more intricate setups encompassing longer option horizons, a greater number of skills, or visual domains. We elaborate on the scalability of ODPP as follows.

(a) The skill horizon is constrained by the MAP inference and eigen decomposition operations previously described. Given their time complexity, the skill horizon could readily be expanded from 50 to 100 or even 500. This augmentation would necessitate an additional time of $\mathcal{O}(10^{-3})$ or $\mathcal{O}(10^{-2})$ seconds per training iteration, compared with previous variational methods. These estimations are based on computations on a machine with a single Intel i7 CPU and four GeForce RTX 2060 GPUs. Note that a skill horizon longer than 100 is rarely necessary. Employing a skill with an excessively long horizon may compromise flexibility in decision-making.

(b) Compared with variational methods, our algorithm does not introduce extra limitations on the number of learned skills. Moreover, in Figure 5(c), we show that even when learning a large number of options at the same time (as much as 60), we can still get options with high quality (mean) and diversity (standard deviation) which increase during the training process.

(c) ODPP needs to learn the Laplacian feature embeddings. For visual domains, this process can incorporate a pretrained CNN model as a feature extractor, which serves to convert visual input into feature vectors. Subsequently, the original algorithm can be applied. Applications in visual domains could pose a common challenge for all option discovery algorithms and present an exciting avenue for future research.

## C.6 Important Hyperparameters

First, we introduce the structure of the networks used in our algorithm and the baselines as follows. We use $s\_dim$, $a\_dim$ to represent the dimension of the state space and action space respectively, and use $c\_num$ to represent the number of options to learn at a time, which can be 10, 20, 40 or 60 in our experiments. Also, we use $tanh$ and $relu$ to denote the hyperbolic tangent function and rectified linear unit used as the activation functions, $FC(X, Y)$, $BiLSTM(X, Y)$ to denote the fully-connected and bidirectional LSTM layer with the input size $X$ and output size $Y$.

- The prior network $P_\omega$ is used in all the algorithms other than DCO and its structure is $[FC(s\_dim, 64), \ tanh, \ FC(64, 64), \ tanh, \ FC(64, 64), \ tanh, \ FC(64, c\_num)]$. The value network corresponding to $P_\omega$ has the same structure as $P_\omega$, except that the output is of size 1.

- The policy network $\pi_\theta$ is used in all the algorithms, with the structure $[FC(s\_dim + c\_num, 64), \ tanh, \ FC(64, 64), \ tanh, \ FC(64, 64), \ tanh, \ FC(64, a\_dim), \ tanh]$. Its corresponding value network has the same structure except that the output is of size 1 and there is not $tanh$ at the end.

- The decoder $P_\phi$ used in ODPP and VALOR takes a sequence of states in the trajectory as input, so it uses bidirectional LSTM as part of the network, i.e., $[BiLSTM(s\_dim, 64), FC(2*64, c\_num)]$. While, the decoder of VIC and DIAYN takes one state as input rather than sequential data, so it uses the fully-connected layer instead, i.e., $[FC(s\_dim, 180), \ tanh, \ FC(180, 180), \ tanh, \ FC(180, 180), \ tanh, \ FC(180, c\_num)]$.

- The option selector $P_\psi$ has the same structure as $P_\omega$ and is used in all the algorithms.

- The eigenfunction network introduced in Section C.3 is used in ODPP and DCO to estimate the Laplacian spectrum. Its structure is $[FC(s\_dim, 256), \ relu, \ FC(256, 256), \ relu, FC(256, 256), \ relu, \ FC(256, 256), \ relu, FC(256, 30)]$, where 30 denotes the dimension of the feature vector $\overrightarrow{b_i}$ mentioned in Section 3.2.

As noted in the paper, the crucial hyperparameters are $\beta$, $\alpha_{1:3}$ in Eq. (6) and (10) which control the importance of each objective term, relating to diversity and coverage. Conducting a grid search on the set of parameters can be exhaustive. Therefore, we follow the process of the ablation study shown in Figure 2, add objective terms and adjust their corresponding weights one by one. In particular, in Figure 2(e), we retain only the $\mathcal{L}^{IB}$ objective and select its weight $\beta = 10^{-3}$ from five possible choices: $1, 10^{-1}, 10^{-2}, 10^{-3}, 10^{-4}$, guided by the visualization results. Next, for Figure 2(f), we introduce $\mathcal{L}_1^{DPP}$ and fine-tune the corresponding weight $\alpha_1$ while keeping $\beta$ fixed at $10^{-3}$. Last, we incorporate $\mathcal{L}_2^{DPP}$ and $\mathcal{L}_3^{DPP}$ and adjust $\alpha_2$ and $\alpha_3$ accordingly, while keeping $\beta$ and $\alpha_1$ fixed. Note that the final two terms must work in tandem to ensure that the discovered options exhibit diversity across different options and consistency for a specific option choice.

After fine-tuning, we set $\beta = 10^{-3}, \alpha_1 = 10^{-4}, \alpha_2 = 10^{-2}, \alpha_3 = 10^{-2}$. It is worthy noting that our evaluations across various challenging RL tasks utilize the same hyperparameter set, highlighting the robustness of our algorithm. This is because our proposed (DPP-based) coverage and diversity measures are task-agnostic and universally applicable to RL tasks.

# D  Additional Evaluation Results

## D.1  Complementary Results on the Effect of the Prior Network

As discussed in Section 3, we learn a prior network $P_\omega$ concurrently with the option policy network $\pi_\theta$. Figure 7 demonstrates how initializing the option selector $P_\psi$ with $P_\omega$ can lead to further performance improvement in the downstream task, using the Point Corridor goal-achieving task as an example. This is based on the fact that both networks share the same structure.

First, in (a), we sample 10,000 trajectories for each option and visualize the agent orientation distribution corresponding to different options. In (b), we present the coordinate system setup along with the four turning points for evaluation. The option choices at these turning points provided by the prior network are displayed in (c). It can be observed that $P_\omega$ favors the most significant options at a given state. For instance, at Location #1, Option #5, which tends to go left or down as shown in (a), is preferred, while at Location #3, Option #3 is favored, guiding the agent to go up or right. Lastly, in (d), we demonstrate that initializing the option selector with the prior network can further enhance the agent's performance in the downstream goal-achieving task.

Previous variational methods choose to fix the prior distribution to avoid a collapse for the prior network to sample only a handful of skills. Our algorithm pretends that because our algorithm additionally introduces three DPP-based terms to explicitly maximize the coverage and diversity of the learned options. With $\mathcal{L}_{1:3}^{DPP}$, each option is expected to cover multiple state clusters with a long range and different options tend to visit different regions in the state space. In this case, the prior network would tend to select multiple diverse skills to improve its learning objective (i.e., Eq.

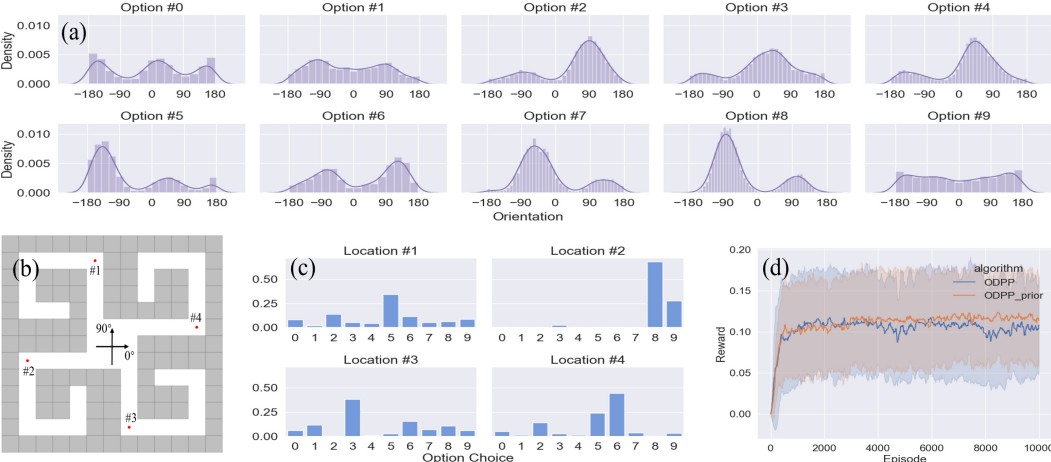

Figure 7: (a) Agent orientation distribution corresponding to different options. (b) Setup of the coordinate system and start points. (c) The output distribution of the prior network at different start points. (d) Performance improvement in the downstream task when applying the prior initialization. The trained prior gives preference to more useful options at corresponding states. At Location #1, Option #5, which tends to go left or down, is preferred; at Location #3, Option #3 is preferred which can lead the agent to go up or right. Moreover, Option #8 is preferred at Location #2 to lead the agent to go down, while Option #6 is preferred at Location #4 to lead the agent to go up.

(12)) rather than sampling only few of them. The collapse happens because the mutual information objective only implicitly measures option diversity as the difficulty of distinguishing them via the variational decoder and does not model the coverage, as noted in Section 3. This motivates us to introduce explicit diversity and coverage measures as regularizers and enhancement.

## D.2 More Visualization of the Learned Ant Locomotion Behaviors

In Figure 8, we show more visualization results of the learned Ant locomotion behaviors without the supervision of task-specific rewards.

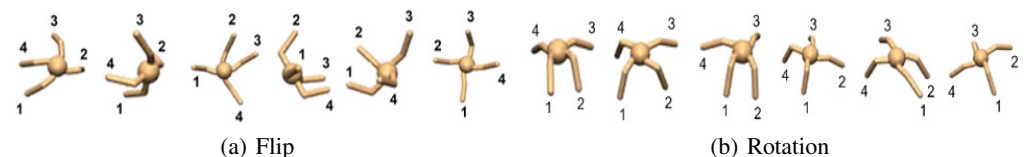

(a) Flip          (b) Rotation

Figure 8: (a) The Ant agent learns to flip over first, then tries to flip back, and finally stands on its Leg 1. (b) The Ant agent walks to the right while rotating. It uses Leg 2&3 as the front legs at first and Leg 1&2 as the front leags at last.

### D.3 Quantitative Ablation Study Results

| Algorithm | Distance | Standard deviation in $x$ | Standard deviation in $y$ |
|---|---|---|---|
| ODPP ($\mathcal{L}^{IB}$) | $9.600 \pm 0.724$ | $5.342 \pm 0.521$ | $9.239 \pm 0.689$ |
| ODPP ($\mathcal{L}^{IB}$, $\mathcal{L}_1^{DPP}$) | $11.037 \pm 0.627$ | $7.067 \pm 0.565$ | $9.313 \pm 0.731$ |
| ODPP ($\mathcal{L}^{IB}$, $\mathcal{L}_{1:3}^{DPP}$) | $\mathbf{14.241 \pm 0.653}$ | $\mathbf{7.493 \pm 0.547}$ | $\mathbf{10.077 \pm 0.799}$ |

Table 1: Numerical results on Mujoco Corridor. We compare our algorithm (the third row) with its ablated versions. ODPP ($\mathcal{L}^{IB}$) represents a version using only the variational objective $\mathcal{L}^{IB}$, whereas ODPP ($\mathcal{L}^{IB}$, $\mathcal{L}_1^{DPP}$) includes the additional coverage objective $\mathcal{L}_1^{DPP}$. For each algorithm, we collect ten different option trajectories and compute the mean distance (i.e., $\sqrt{x^2 + y^2}$) of the final states as the (single-option) coverage measure, and the standard deviation on $x$ and $y$ (of the final states) as the diversity measure. For goal-achieving tasks, these are reasonable metrics, as adopted in VALOR. The table displays the 95% confidence intervals for these metrics from ten repeated experiments. It can be observed that the performance in coverage and diversity improves with the introduction of DPP-based objectives, supporting our algorithm design.

| Algorithm | Distance | Standard deviation in $x$ | Standard deviation in $y$ |
|---|---|---|---|
| ODPP ($\mathcal{L}^{IB}$) | $11.737 \pm 0.581$ | $7.424 \pm 0.450$ | $9.637 \pm 0.766$ |
| ODPP ($\mathcal{L}^{IB}$, $\mathcal{L}_1^{DPP}$) | $13.008 \pm 0.786$ | $8.649 \pm 0.887$ | $9.768 \pm 0.633$ |
| ODPP ($\mathcal{L}^{IB}$, $\mathcal{L}_{1:3}^{DPP}$) | $\mathbf{15.349 \pm 0.629}$ | $\mathbf{10.551 \pm 0.558}$ | $\mathbf{11.356 \pm 0.515}$ |

Table 2: Numerical results on Mujoco Room. These results are gathered in the same way as described above, and the same conclusion can be derived from this table.

### D.4 Baseline Performance on OpenAI Gym and Atari Games

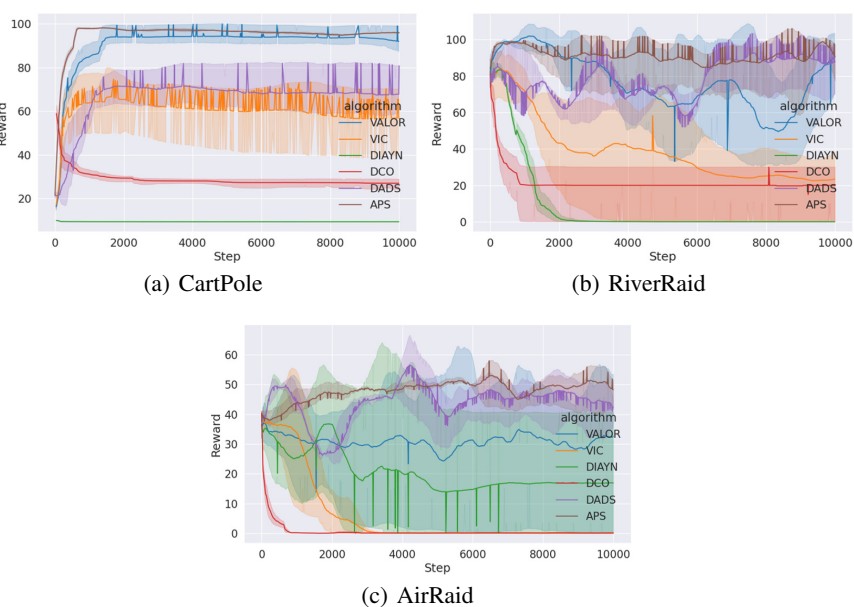

(a) CartPole

(b) RiverRaid

(c) AirRaid

Figure 9: Each algorithm is employed to learn skills for various tasks in an unsupervised manner. The 95% confidence intervals of the option trajectory rewards from repeated experiments with different random seeds are depicted. Comparisons between DADS (brown) and APS (purple) with other baselines illustrate their enhanced performance.

