# OpenReview forum: "A Unified Algorithm Framework for Unsupervised Discovery of Skills based on Determinantal Point Process"
_NeurIPS.cc/2023/Conference — NeurIPS 2023 poster_

### Official Review · Reviewer_6F6e · 2023-07-06

**Soundness:** 3 good
**Presentation:** 2 fair
**Contribution:** 2 fair
**Rating:** 6
**Confidence:** 4

**Summary:**

This paper focuses on unsupervised option discovery. It uses the framework of Determinantal Point Processes (DPP) with the aim of combining the advantages of variational and Laplacian-based methods, and unify the desiderata of coverage and diversity of the learned options. Empirical validation shows the benefits of the approach over baselines in continuous control tasks.

**Strengths:**

- The idea of adopting DPP for option discovery is novel and quite interesting. Ensuring coverage and diversity has been a challenging open question in the unsupervised RL community and the use of DDP is a mathematically clever way to tackle it. The fact that the proposed approach seeks to capture advantages of both variational and Laplacian-based methods, usually part of distinct streams of research, also has a unifying value.
- The paper is quite clear, well motivated and well written.
- Experiments show improvements over baselines like DIAYN.

**Weaknesses:**

- Some related work is missing, including more recent and improved variational methods, such as [1-6]. It would in particular be valuable to discuss the connections with option-discovery approaches like [6-8] that build on the idea of Maximum Entropy Exploration and explicitly seek to optimize coverage, since the latter is a key feature targeted in the paper; see also my first question below.
- The proposed method is quite complicated, with somewhat ad hoc design. While I appreciate the effort to isolate each desiderata (coverage/diversity, intra/inter-option), the final objective function in Eq.(9) is rather involved. In particular, the choice of hyperparameters to trade-off each term looks non-trivial, as acknowledged in the last section of the paper. The method seems quite computationally involved compared to standard variational approaches, and it is unclear how it would scale to more complex setups (e.g, longer horizon, more skills, visual domains).
- The empirical part of the paper could be improved. There have been numerous option-discovery works that clearly outperform VIC/DIAYN and it would be relevant to compare to such stronger baselines. For example: DISDAIN [5], EDL [7] (with SMM [6] instead of the oracle version, despite the fact that it is a 'trajectory-first' method as categorized in the paper), DADS (despite the author's argument of distinction between model-free / model-based). As a small side note, it's also nice to add the random policy as a 'free' baseline (as it sometimes outperforms VIC/DIAYN in coverage objectives).
- Very minor typos: 'extracts' (l.51), 'generalizes the" (l.202), 'long-horizon' (l.251)).

[1] Fast task inference with variational intrinsic successor features, Hansen et al., ICLR 2020

[2] Relative variational intrinsic control, Baumli et al., AAAI 2021

[3] Entropic desired dynamics for intrinsic control, Hansen et al., NeurIPS 2021

[4] Direct then diffuse: Incremental unsupervised skill discovery for state covering and goal reaching, Kamienny et al., ICLR 2022

[5] Learning more skills through optimistic exploration, Strouse et al., ICLR 2022

[6] Active Pretraining with Successor Features, Liu and Abbeel, ICML 2021

[7] Efficient Exploration via State Marginal Matching, Lee et al., arXiv 2019

[8] Explore, discover and learn: Unsupervised discovery of state-covering skills, Campos et al., ICML 2020

**Questions:**

- Could the authors provide a rigorous/mathematical definition of what they mean by 'coverage' throughout the paper. Is it in terms of entropy of states visited by the options H(S)? If so, then it's a term explicitly optimized by diversity I(S,Z) = H(S) - H(S|Z), which would make confusing the statement that "variational option discovery maximizes the diversity of the options through a mutual information loss (while ignoring coverage)" (or maybe the authors mean that optimizing MI exactly is difficult and the approximations performed in the literature tend to have poor coverage, which is different from arguing that the targeted objective ignores coverage in the first place). Also, using a Laplacian-based idea that reduces the cover time of the graph induced by the random policy may be a poor proxy of state space coverage in the entropic sense, while some diversity-based works like [6-8] explicitly target this component with good empirical results. As for the coverage visualizations, Figure 1) b) is said to improve coverage but the options are all going to the same corridor, which does not look desirable in terms of state space coverage. Meanwhile on Figure 3, the paper implies that options learned by (c) are more "diverse" and "beneficial for various downstream tasks" than (a) or (b), although one may argue otherwise, as the latter show less clutter around the initial state and attain more distant regions of the state space. Thanks for the clarifications.
- Isn't the evaluation on Appendix D.3 on OpenAI Gym rather than Atari? (Is Cartpole an Atari game?)
- Although there is a computational complexity discussion in Appendix C.4, could you elaborate more on the scalability of the approach and for example provide numbers on its computational requirements compared to VIC/DIAYN given the same number of environment interactions?
- As first observed in the DIAYN paper, fixing the prior distribution (over option choice at the initial state) rather than learning it prevents a collapse to sampling only a handful of skills (as it can occur for VIC), and this has become standard practice in most subsequent works (although it is a loose lower bound on the original MI objective that should also optimize for the prior distribution). Here you explicitly learn the prior distribution, which is an interesting but undiscussed choice, could you explain more if, and why, you can overcome the aforementioned collapse?

**Limitations:**

A limitation is discussed in Section 5. Not applicable regarding potential negative societal impact.

---

> ### Author Rebuttal · Authors · 2023-08-09
>
> ## Regarding the definition of option coverage (Question #1):
>
> In our paper, coverage is a property defined w.r.t. a single option. It refers to the expected number of landmark states (i.e., clusters of states) traversed by an option trajectory. It is defined as $f(\tau)$ in Eq. (6). We view states in a trajectory as the universal set $\mathcal{X}$. The expected cardinality of a set sampled from $\mathcal{X}$ under a DPP, i.e., $f(\tau)$, reflects the number of modes (i.e., landmark states) contained in the trajectory.
>
> By (a) maximizing coverage of each single option and (b) maximizing diversity among different options in the meanwhile, the overall span of all options in the state space can be maximized. The "coverage" in our paper does not refer to the span of all options and so we do not use entropy of states as the coverage measure.
>
> In Figure 1(b), options optimized for (single-option) coverage all enter the right corridor, allowing each one’s trajectory to visit more landmark states and thus achieving better (single-option) coverage compared with the ones in Figure 1(a). In Figure 3(a)(b), we show the learned option trajectories starting from different locations (i.e., yellow points), while in 3(c), we only show trajectories starting from the center point. Options learned in (c) can lead the agent to multiple directions from a starting point, showing more diversity.
>
> ## Regarding complexity and scalability of ODPP (Weakness #2, Question #3):
>
> Please refer to the global response at the top.
>
> ## Regarding the prior network (Question #4):
>
> Learning a prior network provides a tighter lower bound on the original MI objective. In Figure 6 of Appendix D.1, we show that the learned prior network can be used as initializations to aid downstream tasks.
>
> Previous variational methods choose to fix the prior distribution to avoid a collapse for the prior network to sample only a handful of skills. Our algorithm pretends that because our algorithm additionally introduces three DPP-based terms to explicitly maximize the coverage and diversity of the learned options. With $\mathcal{L}^{DPP}_{1:3}$, each option is expected to cover multiple state clusters with a long range and different options tend to visit different regions in the state space. In this case, the prior network would tend to select multiple diverse skills to improve the learning objective (i.e., Eq. (11)) rather than sampling only few of them.
>
> The collapse happens because the mutual information objective only implicitly measures option diversity as the difficulty of distinguishing them via the variational decoder and does not model the coverage, as noted in Section 3. This motivates us to introduce explicit diversity and coverage measures as regularizers and enhancement.
>
> ## Regarding related works (Weakness #1):
>
> Here, we provide comparisons of our algorithm with more recent variational option discovery methods. (This is a simplified version due to the word count limit. We can provide the detailed version in the discussion stage (if asked) and the final submission. [1-8] refers to papers listed by the reviewer.)
>
> In  [1], the authors focus on alternative approaches to leverage options learned during the unsupervised phase, rather than new option discovery algorithms. They still adopt objectives based on Mutual Information (MI) as previous works without solving the exploration issue.
>
> The authors of [2] propose a slightly-modified version of VIC to improve usefulness of the discovered options by introducing an extra posterior. Still, their options are not explicitly trained for better coverage/exploration.
>
> The authors of [3] propose to replace the fixed prior distribution $P(c)$ with a fixed dynamics model over the option latent codes $P(c_{t}|c_{t-1})$. Each latent code corresponds to a sub-trajectory. By concatenating sub-trajectories, the agent can reach much further states. They only rely on MI-based objectives, so they cannot model the coverage of each option (like ours) and instead choose to chain options for better overall coverage.  The fixed $P(c_{t}|c_{t-1})$ can result in inflexibility when applying options in downstream tasks.
>
> In [4], they employ a multi-step protocol to generate options organized in a tree-like structure. Heuristics and structural limits are involved in each step, which may hinder its generality. Also, they propose to optimize the local coverage around the final state rather than the overall trajectory coverage like ours.
>
> In [6], they optimize the MI $I(s, c) = H(s) - H(s|c)$, where $H(s)$ is for improving exploration of the learned options. They adopt a variational posterior $P_{\phi}(s|c)$ to estimate $H(s|c)$. Learning $P_{\phi}(s|c)$ can be challenging when the state space is high-dimensional, compare with $P_{\phi}(c|s)$ used in our paper, which can hinder the optimization.  Further, in DADS, they categorize algorithms utilizing $I(s, c) = H(s) - H(s|c)$ as the forward form of MI-based option discovery, and they empirically and theoretically show the limited capability of these algorithms for exploration even with $H(s)$ in the objective.
>
> EDL [8] disentangles exploration and skill discovery into two phases. They first train an exploration policy that induces a uniform distribution over states and can cover the state space. Then, they discover diverse skills contained in thorough samples from the pre-learned exploration policy, based on MI-based objectives. Training such an exploration policy can be challenging. In [8], they adopt SMM [7] as a solution, which requires solving a demanding max-min problem. In our paper, we tackle a more challenging scenario where the agent must learn to identify diverse options and thoroughly explore the environment at the same time, starting from a random policy, without access to expert trajectories or exploration policies.
>
> ## Regarding Weakness #3:
>
> We have offered comparisons with more advanced baselines: DADS and APS [6], in the global response as a PDF.

---

> > ### Comment · Reviewer_6F6e · 2023-08-18
> >
> > Thanks for the rebuttal which addresses most of my concerns and questions. Incorporating the clarifications, related work and new empirical results will improve the paper. The proposed method, if somewhat complicated, brings interesting insights to diversity and coverage in unsupervised option discovery. I have raised my score accordingly.

---

> > > ### Author Response · Authors · 2023-08-18
> > >
> > > Thank you for the raise. We greatly appreciate it. we will add our rebuttal content to the final submission, including clarifications on the definition of coverage, analysis on the complexity and scalability of ODPP, related works, and comparisons with more advanced baselines.

---

### Official Review · Reviewer_vq5Y · 2023-07-07

**Soundness:** 3 good
**Presentation:** 3 good
**Contribution:** 3 good
**Rating:** 7
**Confidence:** 3

**Summary:**

This paper introduces a novel framework for unsupervised option discovery by utilizing Determinantal Point Process (DPP) to quantify and optimize both the diversity and the coverage of the learned options. The proposed unified option discovery framework captures the advantages of both variational and Laplacian-based methods, which is the major tools for existing unsupervised option discovery approaches. The experiment results in both MuJoCo and Atari demonstrate the superiority of the proposed algorithm.

**Strengths:**

1. The motivation to propose the unified algorithm framework is convincing and the paper is well-written and easy to follow. The author well illustrates the main idea of this paper using a toy example.
2. The proposed option discovery framework unifies both variational and Laplacian-based methods and enables explicit maximization of diversity and coverage of the options.
3. Though DPP is widely used in methods to promote diversity, it is the first work to adopt it into option diversity.
4. Both the theoretical and experimental results show the superiority of the algorithms.

**Weaknesses:**

1. The major concern to me is the application of the proposed framework. In Equ 9. there are three hyper-parameters in the loss function. How to tune these hyper-parameters?  In L586 you say fine-tune important hyperparameters using a sequential, greedy method based on options’ visualization, such as in Fig2. Could you provide more details?
2. DPP is widely used in the work to promote diversity. It would be better to discuss them in related work, e.g., in promoting diverse policies in population-based RL[1, 2], diverse policies in games [3], recommendation diversity [4], etc.
3. Some theoretical results can be highlighted (such as using a Proposition station) in the main text.
4. In Sec4, instead of some visualization results, are there some diversity metrics that could be proposed to measure the diversity? It would be better to see these numerical results in the main text.

[1] Jack Parker-Holder, Aldo Pacchiano, Krzysztof M Choromanski, and Stephen J Roberts. Effective diversity in population based reinforcement learning. Advances in Neural Information Processing Systems, 33:18050–18062, 2020.

[2] Wu S, Yao J, Fu H, et al. Quality-Similar Diversity via Population Based Reinforcement Learning. In The Eleventh International Conference on Learning Representations, (2023).

[3] Perez-Nieves, Nicolas, et al. "Modelling behavioural diversity for learning in open-ended games." International conference on machine learning. PMLR, 2021.

[4]. Chen, L., G. Zhang, E. Zhou. Fast greedy MAP inference for determinantal point process to improve recommendation diversity. In Advances in Neural Information Processing Systems 31, NeurIPS 2018, pages 5627–5638. 2018.



**Questions:**



**Limitations:**

---

> ### Author Rebuttal · Authors · 2023-08-09
>
> ## Regarding fine-tuning the hyperparameters:
>
> As noted in the paper, the crucial hyperparameters are $\beta,\ \alpha_{1:3}$ in Eq. (4) and (9) which control the importance of each objective term, relating to diversity and coverage. Conducting a grid search on the set of parameters can be exhaustive. Therefore, we follow the process of the ablation study shown in Figure 2, add objective terms and adjust their corresponding weights one by one. In particular, in Figure 2(e), we retain only the $\mathcal{L}^{IB}$ objective and select its weight $\beta=10^{-3}$ from five possible choices: $1, 10^{-1}, 10^{-2}, 10^{-3}, 10^{-4}$, guided by the visualization results. Next, for Figure 2(f), we introduce $\mathcal{L}^{DPP}_1$ and fine-tune the corresponding weight $\alpha_1$ while keeping $\beta$ fixed at $10^{-3}$. Last, we incorporate $\mathcal{L}^{DPP}_2$ and $\mathcal{L}^{DPP}_3$ and adjust $\alpha_2$ and $\alpha_3$ accordingly, while keeping $\beta,\ \alpha_1$ fixed. Note that the final two terms must work in tandem to ensure that the discovered options exhibit diversity across different options and consistency for a specific option choice.
>
> After the fine-tuning, we set $\beta=10^{-3}, \alpha_1=10^{-4}, \alpha_2=10^{-2}, \alpha_3=10^{-2}$. It is worthy noting that our evaluations across various challenging RL tasks utilize the same hyperparameter set, highlighting its robustness. This is because our proposed (DPP-based) coverage and diversity measures are task-agnostic and universally applicable to RL tasks.
>
> ## Regarding related works on DPP:
>
> We will add the following literature review on applying DPP to diversity enhancement:
>
> Determinantal Point Processes (DPPs) have found applications across a wide array of domains to promote diversity due to their unique ability to model diverse subsets. Originating from quantum physics, DPPs were introduced to the machine learning community by Kulesza and Taskar [2], whose tutorial highlighted the potential of DPPs for promoting diversity. In information retrieval tasks, Wilhelm et al. [10] and Chen et al. [11] have demonstrated the utility of DPPs in diversifying the output of recommendation systems.  In Computer Vision, Gong et al. [3] and Kim et al. [4] utilized DPPs in video summarization and object detection, respectively, to reduce outcome redundancy. In Natural Language Processing, Perez-Beltrachini et al. [5] exploited DPPs to select relevant and diverse content for neural abstractive summarisation, and Song et al. [6] applied them to model the query-level and system-level diversity in neural conversation systems. Expanding on this theme of diversity, DPPs have been extensively applied in reinforcement learning, particularly in promoting diverse policies in population-based RL [7, 8], and diverse policies in games [9].
>
> ## Regarding the theoretical results:
>
> Thanks for your advice. We will highlight the variational lower bound of the Information Bottleneck objective (i.e., Eq. (5)) and the unbiased gradient estimators for the prior network and intra-option policy (i.e., Eq. (10)-(12)), and formalize them as propositions in the final version.
>
> ## Regarding the diversity metrics:
>
> In Figure 5(b), we utilize the standard deviation of trajectory rewards corresponding to different options as quantitative measures for the option diversity. This measure has been used in previous works, such as [1].
>
> For more convincing ablation study, we provide new quantitative results in the global response as a PDF. We propose using the distribution of final locations within option trajectories to measure the diversity and coverage, as in [12].
>
> ## References:
>
> [1] Eysenbach, Benjamin, Abhishek Gupta, Julian Ibarz, and Sergey Levine. "Diversity is All You Need: Learning Skills without a Reward Function." In International Conference on Learning Representations. 2018.
>
> [2] Kulesza, Alex, and Ben Taskar. "Determinantal point processes for machine learning." Foundations and Trends® in Machine Learning 5, no. 2–3 (2012): 123-286.
>
> [3] Gong, Boqing, Wei-Lun Chao, Kristen Grauman, and Fei Sha. "Diverse sequential subset selection for supervised video summarization." NeurIPS, 2014.
>
> [4] Kim, Nuri, Donghoon Lee, and Songhwai Oh. "Learning instance-aware object detection using determinantal point processes." Computer Vision and Image Understanding 201 (2020): 103061.
>
> [5] Perez-Beltrachini, Laura, and Mirella Lapata. "Multi-document summarization with determinantal point process attention." JAIR 71 (2021): 371-399.
>
> [6] Song, Yiping, Rui Yan, Yansong Feng, Yaoyuan Zhang, Dongyan Zhao, and Ming Zhang. "Towards a neural conversation model with diversity net using determinantal point processes." AAAI, 2018.
>
> [7] Parker-Holder, Jack, Aldo Pacchiano, Krzysztof M. Choromanski, and Stephen J. Roberts. "Effective diversity in population based reinforcement learning." NeurIPS, 2020.
>
> [8] Wu, Shuang, Jian Yao, Haobo Fu, Ye Tian, Chao Qian, Yaodong Yang, Qiang Fu, and Yang Wei. "Quality-Similar Diversity via Population Based Reinforcement Learning." ICLR, 2022.
>
> [9] Perez-Nieves, Nicolas, Yaodong Yang, Oliver Slumbers, David H. Mguni, Ying Wen, and Jun Wang. "Modelling behavioural diversity for learning in open-ended games." ICML, 2021.
>
> [10] Wilhelm, Mark, Ajith Ramanathan, Alexander Bonomo, Sagar Jain, Ed H. Chi, and Jennifer Gillenwater. "Practical diversified recommendations on youtube with determinantal point processes." CIKM, 2018.
>
> [11] Chen, Laming, Guoxin Zhang, and Eric Zhou. "Fast greedy map inference for determinantal point process to improve recommendation diversity." NeurIPS, 2018.
>
> [12] Achiam, Joshua, Harrison Edwards, Dario Amodei, and Pieter Abbeel. "Variational option discovery algorithms." arXiv:1807.10299 (2018).

---

> > ### Comment · Reviewer_vq5Y · 2023-08-15
> >
> > Thanks for your response. I am happy that the author solved most of my questions. It would be nice to add these details to the future version.
> > I am happy to raise my score.

---

> > > ### Author Response · Authors · 2023-08-15
> > >
> > > Thank you for the raise. We greatly appreciate it. We will add our rebuttal content to our final submission, including details on the fine-tuning process, related works on DPP, new quantitative results on the ablation study, and more formalized theoretical results.

---

### Official Review · Reviewer_iqV6 · 2023-07-08

**Soundness:** 3 good
**Presentation:** 3 good
**Contribution:** 3 good
**Rating:** 6
**Confidence:** 3

**Summary:**

The paper introduces an unsupervised option discovery framework based on a combination of Determinantal Point Process and Laplacian spectral features. The main idea is to combine variational methods and  Laplacian methods in order to control for coverage and diversity of the options. The proposed method has been validated experimentally on Mujoco 3D navigation environments where it shows superior performance to other approaches. Further experiments have been done on Atari environments as well.

**Strengths:**

The proposed combination of DPP and Laplacian features is reasonable and shows promising results. The novelty stems mainly from the modified DPP kernel similarity matrix which makes use of the Laplacian spectrum. I also think that maximizing the cardinality of the the landmark set as a diversity signal is a nice idea.

All components that were introduced in the method are well argued. Especially the overarching motivation of maximizing coverage and diversity in the same time (reaping the benefits of variational and Laplacian methods)

Experiments are convincing.

**Weaknesses:**

At some points the paper lacks clarity, for example in the exposition of the DPP, terms were introduced that were not properly explained and that I would not consider common knowledge.

Some probabilistic framing that was used in the paper was off, but we can clarify this in the rebuttal phase hopefully.

The Atari experiments should appear in the main paper in some form, since they are already announced in the abstract.

As the authors have already noted in the paper, the method needs balancing of 3 terms in the objective which might make it impractical, however the hyperparameters are stable across different environments (shows robustness).

No significant theoretical insights, rather a combination of existing works in order to devise and option discovery algorithm.

The writing in general could be improved, I found it a bit hard to keep track of all the introduced terms and the connection to the main idea of the paper.

**Questions:**

line 112: the explanation of this Gram matrix seems lacking, for instance the quality measure is just introduced in the text here and used once more later, without any explanation how it related to reward and the DPP probability of a subset.

eq.3: it is not clear to me from this equation that you are doing MAP inference,  can you specify in clear terms what is the posterior here and what is the prior (also in the paper)?

eq. 4: what is being maximized over here? And a follow-up, in which sense is beta a Lagrange multiplier? What is the constrained optimization problem that is being solved here, can you actually write it this way? Please clarify this in your response and ideally in the main text.

line 219: the notation here is confusing, is this to define a conditional random variable given s_0 and c? (the trajectory). Wouldn’t it be better to just define this as sampled from a conditional distribution, then take the expectation in eq.7 over it. (might be less confusing)

eq.8: same comment as for line 219.

eq.9: perhaps you should consider putting the minus sign into the second loss, so that it remains a proper loss (something that you want to minimize) and adjust the explanation.

figure 3 caption - “in the normal setting”

the legends in figure 4 are not well visible and the fonts are off. I suggest you place the legends outside of the figure.


**Limitations:**

Yes

---

> ### Author Rebuttal · Authors · 2023-08-09
>
> ## Regarding the Gram Matrix (Question #1):
>
> As introduced in Section 2.2, the Gram Matrix includes the quality measures $q$ and normalized vectors $\vec{b}$ for each element in the set. From Eq. (1), we can see that the sampling probability is proportional to the squared volume of the parallelepiped spanned by the columns $q_i\vec{b}_{i}$, for $i$ in the sampled subset. Thus, elements with more orthogonal feature vectors and higher quality values are more probable to be sampled as a subset.
>
> In RL tasks, we need to assign quality measures and features for each state. States with higher expected returns should be visited more frequently and thus be assigned with higher quality values. However, in our reward-free setting, we do not have prior knowledge on the quality of states, so we define equal quality measures to each state as 1, as mentioned in Section 3.2.
>
> ## Regarding Eq. (3) (Question #2):
>
> $P_{L(\mathcal{W})}(\mathbb{W}=W)=P(\mathbb{W}=W|\mathbb{L}=L(\mathcal{W}))$ denotes the probability of sampling the subset $W$ out of the universal set $\mathcal{W}$ given the definition of the kernel matrix $L(\mathcal{W})$. With such a conditional model, the problem to find the set $W \subseteq \mathcal{W}$ with the highest probability, i.e., Eq (3), is referred to as maximum a posteriori (or MAP) inference, as defined in the first paragraph of Section 2.4.5 in [1]. We will clarify this in the main text during the final submission stage.
>
> ## Regarding Eq. (4) (Question #3):
>
> Our goal is to learn a policy $\pi_{\theta}$ and a prior $P_{\omega}$ which condition on the option choice $c$. $c$ should be maximally expressive about the landmark states $G$ induced by $\pi_{\theta}$ and $P_{\omega}$, while being compressive about the whole trajectory $\tau$ to eliminate redundant information. According to the Information Bottleneck framework [2], this can be realized through:
>
> $\max_{\theta, \omega} \mathbb{E}_{s_0 \sim \mu(\cdot)} I(c, G|s_0;\theta, \omega)$
>
> $I(c, \tau|s_0;\theta, \omega) \leq I_{ct}$
>
> where $I_{ct}$ is the information constraint. Equivalently, with the introduction of a Lagrange multiplier $\beta \geq 0$, we can optimize: $\max_{\theta, \omega} \mathbb{E}_{s_0 \sim \mu(\cdot)} \left[ I(c, G|s_0;\theta, \omega) - \beta I(c, \tau|s_0;\theta, \omega) \right]$. Thanks for the advice. We will add this to the main text during the final submission stage.
>
> ## Regarding Eq. (7)-(9):
>
> $\vec{\tau}_{(s_0, c)}$ denotes a set of $M$ sampled trajectories subject to the option choice $c$ and starting from $s_0$. We will adopt your suggested modifications to Eq. (7) - (9) for better explanation.
>
> ## Regarding Figure 3 and 4:
>
> Thank you for your careful check. We will fix them during the final submission stage.
>
> ## References:
>
> [1] Kulesza, Alex, and Ben Taskar. "Determinantal point processes for machine learning." Foundations and Trends® in Machine Learning 5, no. 2–3 (2012): 123-286.
>
> [2] Alemi, Alexander A., Ian Fischer, Joshua V. Dillon, and Kevin Murphy. "Deep Variational Information Bottleneck." In International Conference on Learning Representations, 2017.

---

### Official Review · Reviewer_7oxb · 2023-07-23

**Soundness:** 2 fair
**Presentation:** 3 good
**Contribution:** 3 good
**Rating:** 6
**Confidence:** 3

**Summary:**

This paper addresses reward-free options discovery for RL. First, it notes that prior work would prioritize either state coverage or diversity into the options discovery procedure. Hence, it proposes a new loss function that fosters coverage and diversity simultaneously by exploiting DPPs on both the trajectories generated by the options and the states within a trajectory. Finally, it presents an algorithm to optimize this objective, which is evaluated in continuous control domains against standard baselines, such as VIC, VALOR, DIAYN, and Laplacian methods.

**Strengths:**

- Exploiting tools from DPPs for the options discovery is interesting and, to the best of my knowledge, novel and original;
- The experimental results are at least promising;
- The paper includes a neat presentation of the previous works and approaches for unsupervised options discovery.

**Weaknesses:**

- The learning objective and the corresponding algorithm are quite convoluted. They requires several layers of approximation to make the learning tractable, as well as a handful of hyper-parameters to be tuned and the definition of a suitable kernel over trajectories and states;
- The experimental evaluation is not thorough. Most of all, the ablation study is limited to a single run in a single domain, and leaves the reader wondering whether all of the introduced ingredients would be needed in general;
- The intuition behind the loss function is not always pristine. It is easy to lose focus going through Section 3 of the paper;
- The visualization of the experimental results could be further polished.

**Questions:**

The paper is an interesting contribution to a problem that is somehow far from being solved despite receiving a considerable attention recently. While the main contribution of this paper is algorithmic, it is hard to assess its value without a stronger empirical study (or more theoretical corroboration). Thus, I am currently providing a slightly negative score, even if a case for acceptance could be made and I am open to change my score after the authors' response. As is, the paper looks like a nice engineering feat to me, but lacks some convincing intuition and experiments on why all those tricks would work in general.

**Algorithm**

- There is one aspect that is not completely clear to me on how the loss is presented. The authors motivates the work as going beyond previous variational or Laplacian approaches with a new framework that comprises both, but then the main ingredient of the loss, i.e., $\mathcal{L}_{IB}$, looks a standard variational loss with some variation (landmark states). Moreover, the components $\mathcal{L}_1, \mathcal{L}_2, \mathcal{L}_3$ already provide incentives for coverage and diversity, so why one should also add the information bottleneck on top of them? Overall, I think that presenting this work as an evolution of variational option discovery would help its presentation.
- From my understanding, to define the relevant DPPs a proper kernel on trajectories and states is needed. Can the authors discuss how this would be designed in the absence of any domain knowledge?
- I am wondering whether the presented loss function induces a (hidden) RL problem or not, i.e., whether the loss function could be incorporated into a standard reward function in hindsight. Is the approach producing a set of deterministic options or the discovered options may be stochastic?

**Experiments**

- The presented approach is fairly complicated and I think the ablation study in Section 4.1 is not enough to state that all of the introduced ingredients would benefit the option discovery in general. It is somehow underwhelming that the paper only presents one result, a single domain and a single seed, through an illustration instead of some quantitative analysis and learning curves. I believe this does not meet the bar to motivate the added complexity of the loss function, especially on top of the $\mathcal{L}_{IB}$ that seems to lead to a good options discovery already.
- All of the approach is predicated on the need for coverage and diversity simultaneously, but it is hard to evaluate them beyond qualitative assessments. However, I think the paper could put more effort in designing quantitative measures to evaluate diversity and coverage.
- I would suggest to report the learning curves with average and confidence intervals instead of one standard error, which is hardly meaningful.

**Minor**

- I would suggest to report the pseudocode of Algorithm 1 in the main text.
- The overall notation, in particular the one needed to define the loss function, is sometimes convoluted.
- Where is the DIAYN learning curve in Fig. 4b, 4c?
- Why are the Atari results relegated to the appendix? CartPole is not an Atari game, but a continuous control task.
- In the downstream task setting only the option selecting policy is learned? Do the authors considered also fine-tuning of the learned options?

**Limitations:**

The paper explicitly addresses the limitations of the presented approach in the final paragraph.

---

> ### Author Rebuttal · Authors · 2023-08-09
>
> ## Regarding loss terms (Question #1):
>
> As noted in the first paragraph of Section 3, we need to learn an intra-option policy $\pi_{\theta}(a|s,c)$ conditioned on the option choice $c$. Each option choice should correspond to a specific policy. As a common practice in variational methods, this type of option-policy mapping is established by maximizing the mutual information between them, i.e., $\mathcal{L}^{IB}$ in our case. The DPP-based loss terms, i.e., $\mathcal{L}^{DPP}_{1:3}$, cannot be harnessed directly to obtain such a mapping.
>
> Presenting our algorithm as an evolution of variational methods would be a good idea. We provide such a view in the second paragraph of Section 3 as an algorithm overview. The variational loss term, $\mathcal{L}^{IB}$, is used to establish the option-policy mapping, so we can learn multiple options simultaneously by introducing a conditional variable $c$. However, the mutual information objective only implicitly measures diversity of options as the difficulty of distinguishing them via a variational decoder (i.e., $P_{\phi}$ in our paper), and does not model the coverage. As a novel extension, we propose to explicitly model and optimize the coverage and diversity of options based on DPP.
>
> ## Regarding the DPP kernel (Question #2):
>
> To make our algorithm general, we do not rely on domain knowledge when designing the DPP kernel.
>
> For the kernel matrix on the set of states, as described in the second paragraph of Section 3.2, we need to specify the quality measure and feature vector for each state. Since this is a reward-free setting, all states are assigned with the same quality. As for feature vectors, we use Laplacian spectrum (i.e., eigenvectors corresponding to the $D$ smallest eigenvalues) of the state-transition graph, which intuitively captures connectivity among states. The same feature design has been used for spectral clustering [1], and we have shown analytically and empirically that we can generalize Laplacian option discovery with this feature embedding. This feature design is task-irrelevant. In any RL task, we can compile state transitions in replay buffers, upon which we can estimate these features, as detailed in Appendix C.3.
>
> As for the kernel matrix on trajectories, we still assign equal quality measure to each trajectory. As in the first paragraph of Page 6, the feature of each trajectory is acquired as a sum over feature vectors of states within the trajectory. This follows the structural DPP framework [2] used for modeling sequential data. Still, the kernel matrix design does not require domain knowledge.
>
> ## Regarding the option learning (Question #3):
>
> The learning outcome is the intra-option policy $\pi_{\theta}(a|s,c)$. $c$ is a one-hot vector representing a discrete set of options. For each $c$, its policy is a mapping from the current state to the action, which is stochastic in continuous control tasks.
>
> $\pi_{\theta}(a|s,c)$ is modeled as a neural network and trained through applying the gradient $\nabla_{\theta}\mathcal{L}$ (i.e., Eq. (10)). The gradient form inspires us to update the policy with Actor-Critic methods, where $A^{\pi_{\theta}}_m$ defines the Q-function.
>
> When being applied to downstream tasks, $\pi_{\theta}(a|s,c)$ is fixed and we only need to learn a high-level policy $P_{\psi}(c|s)$ to select among options. For each selected option, we execute its intra-option policy for a fixed number of steps (i.e., the option horizon) before sampling a new one.
>
> ## Regarding the experiments:
>
> For quantitative results, we have provided comparisons among multiple option discovery algorithms, including the learning performance on downstream Mujoco tasks (Figure 4), effectiveness of the discovered skills in 3D Mujoco Locomotion (Figure 5) and OpenAI Gym tasks (Figure 8 in Appendix D.3). Besides, we provide detailed analysis on the learned prior network (Figure 6 in Appendix D.1).
>
> The introduction of each objective term is intuitive. $\mathcal{L}^{IB}$ is for establishing the option-policy mapping and implicitly encouraging the diversity, $\mathcal{L}^{DPP}_{1}$ is for improving the coverage of each option trajectory, while the other DPP terms work together to explicitly model and optimize the option diversity.
>
> For more convincing ablation study, we provide new quantitative results in the global response as a PDF. We propose using the distribution of final locations within option trajectories to measure the diversity and coverage, as in [3].
>
> ## Regarding the minor issues:
>
> We will move the pseudocode to the main text since we get an extra page in the final version.
>
> The learning curves of DIAYN in Figure 4(b)(c) are blocked by the ones of other algorithms. The options learned with these algorithms can hardly lead the agent out of the inner room in Figure 4(a) to reach the goal area and get any reward signal.
>
> CartPole is not an Atari game like the other two in Figure 8. We will fix the benchmark name as OpenAI Gym and move Figure 8 to the main text.
>
> In the downstream task setting, only the option selecting policy is learned. We do not fine-tune the options in this stage, because this is for evaluating the discovered options in the previous stage and we need to keep them fixed in downstream tasks to keep comparisons fair.
>
> ## Regarding the complexity of the proposed algorithm (Weakness #1):
>
> We have provided a discussion on this in the global response.
>
> ## References:
>
> [1] Ng, Andrew, Michael Jordan, and Yair Weiss. "On spectral clustering: Analysis and an algorithm." Advances in neural information processing systems 14 (2001).
>
> [2] Kulesza, Alex, and Ben Taskar. "Structured determinantal point processes." Advances in neural information processing systems 23 (2010).
>
> [3] Achiam, Joshua, Harrison Edwards, Dario Amodei, and Pieter Abbeel. "Variational option discovery algorithms." arXiv preprint arXiv:1807.10299 (2018).

---

> > ### Comment · Reviewer_7oxb · 2023-08-19
> > **After response**
> >
> > I am very sorry for the late reply. First, I want to thank the authors for their detailed clarifications, which make me feel now more confident in evaluating this paper. My feeling is that the empirical analysis could be further strengthen, but the overall contribution of this paper is original and interesting. I am raising my score accordingly.

---

> > > ### Author Response · Authors · 2023-08-19
> > >
> > > Thank you for the raise. We greatly appreciate it. We will bolster the analysis of empirical results and include rebuttal content in our final submission, encompassing insights on $\mathcal{L}^{IB}$, clarifications for the DPP kernel and option learning outcomes, complexity analysis of ODPP, and quantitative results for the ablation study.

---

### Official Review · Reviewer_4NNK · 2023-08-03

**Soundness:** 3 good
**Presentation:** 3 good
**Contribution:** 3 good
**Rating:** 7
**Confidence:** 4

**Summary:**

This paper proposes and evaluates an approach to option discovery in reinforcement learning. The aim is to autonomously identify a set of options that are diverse and that give good coverage of the state space. This aim is achieved by using the Determinantal Point Process (DPP). The proposed approach is evaluated in a range of domains.

**Strengths:**

The proposed approach is intuitive and principled.

While the underlying notions of diversity and coverage have been used earlier in the literature for option discovery, their combined optimization through the Determinantal Point Process is novel and returns better results than related existing methods.

The experimental evaluation is extensive and varied, showing not only learning curves but also option trajectories.


**Weaknesses:**

Only minor comments:

An analysis of computational complexity is provided in the supplementary material. It would be useful to see a short summary of that in the main paper.

In the learning curves, it would be useful to plot ceiling performance.

Text in Figure 1d is too small.




**Questions:**

No questions.

**Limitations:**

Adequate.

---

> ### Author Rebuttal · Authors · 2023-08-09
>
> Thank you for your appreciation. We will fix the issues that you mentioned during the final submission stage, including moving analysis of the computational complexity to the main text, highlighting the ceiling performance in the learning curves, and adjusting the text size in Figure 1(d).

---

### Author Rebuttal · Authors · 2023-08-09

## Regarding complexity of ODPP :

The learning target of ODPP is an intra-option policy $\pi_{\theta}(a|s,c)$ conditioned on the option choice $c$. As in Section 3.3, this policy is learned with an Actor-Critic algorithm for which the Q-function is defined as Eq. (12). This Q-function contains variational and DPP-based objectives. Compared with previous variational option discovery algorithms (e.g., DIAYN, VIC, VALOR), we additionally need to (a) sample landmark states from each trajectory and (b) calculate the DPP-related terms: $f(\cdot),\ g(\cdot),\  h(\cdot)$.

For (a), we adopt a fast greedy MAP inference algorithm for DPP [1]. As mentioned in Appendix C.2, it takes $\mathcal{O}(S^2N)$ to sample $S$ landmark states from an option trajectory of length $N$. In our setting, $N=50,\ S=10$, so the process can be done in real-time.

For (b), we need to build DPP kernel matrices, and then compute $f(\cdot),\ g(\cdot),\  h(\cdot)$ based on eigenvalues of the corresponding kernel matrix as in Eq. (6)-(8). The time complexity for eigen decomposition is $\mathcal{O}(N^3)$, where $N$ is the size of the matrix. For the state kernel matrix, $N$ is the number of states in an option trajectory (i.e., the option horizon) which we set as 50. For the trajectory matrix, $N$ corresponds to the number of trajectories collected in each training iteration, which is set as 100. Thus, $f(\cdot),\ g(\cdot),\  h(\cdot)$ can be computed in real-time.

To build the kernel matrix, we need feature vectors for each state. As introduced in Appendix C.3, the feature vector is the output of a pre-trained neural network which takes the state as input. The training of this feature function is based on state transitions in the replay buffer and only needs to be done for once or twice in the whole option discovery process, of which the time cost is within 30 minutes.

To sum up, compared with previous variational methods, we additionally introduce three DPP items to explicitly model the option diversity and coverage, of which the involvement only slightly increases the time complexity.

## Regarding scalability of ODPP:

ODPP can indeed be adapted to more intricate setups encompassing longer option horizons, a greater number of skills, or visual domains. We elaborate on scalability of ODPP as follows and will include these discussions and numerical examples in the revised paper.

(a) The skill horizon is constrained by the MAP inference and eigen decomposition operations previously described. Given their time complexity, the skill horizon could readily be expanded from 50 to 100 or even 500. This augmentation would necessitate an additional time of $\mathcal{O}(10^{-3})$ or $\mathcal{O}(10^{-1})$ seconds per training iteration, compared with previous variational methods. These estimations are based on computations on a machine with a single Intel i7 CPU and four GeForce RTX 2060 GPUs. Note that a skill horizon larger than 100 is rarely necessary. Employing a skill with an excessively long horizon may compromise flexibility in decision-making.

(b) Compared with variational methods, our algorithm does not introduce extra limitations on the number of learned skills. Moreover, in Figure 5(c), we show that even when learning a large number of options at the same time (as much as 60), we can still get options with high quality (mean) and diversity (standard deviation) which increase during the training process.

(c) ODPP needs to learn the Laplacian feature embeddings. For visual domains, this process can incorporate a pretrained CNN model as a feature extractor, which serves to convert visual input into feature vectors. Subsequently, the original algorithm can be applied. Applications in visual domains could pose a common challenge for all option discovery algorithms and present an exciting avenue for future research.

## Regarding new results:

As required by reviewers, new empirical results are provided in the uploaded PDF, including quantitative ablation study results and comparisons with more advanced baselines on OpenAI Gym.

## References:

[1] Chen, Laming, Guoxin Zhang, and Eric Zhou. "Fast greedy map inference for determinantal point process to improve recommendation diversity." Advances in Neural Information Processing Systems 31 (2018).

---

### Decision · Program_Chairs · 2023-09-21

**Decision:**

Accept (poster)

**Comment:**

The paper focuses on the problem of unsupervised option discovery. The authors proposes a novel and effective mathematical formulation based on DPPs to achieve diversity and coverage at the same time. The resulting algorithm successfully improves existing baselines in standard benchmarks.

After the discussion, there is general consensus that the paper investigates an important problem and it introduces a novel formulation that combines existing but often orthogonal approaches to unsupervised skill discovery such as Laplacian and mutual-information-based approaches. The empirical evaluation is overall satisfactory.

I strongly encourage the authors to fully integrate their rebuttal into the final version to properly address concerns related to the method formulation, its complexity, and the experiments.